# A unified platform to manage, share, and archive morphological and functional data in insect neuroscience

Stanley Heinze[1,2]*, Basil el Jundi[3], Bente G Berg[4], Uwe Homberg[5], Randolf Menzel[6], Keram Pfeiffer[3], Ronja Hensgen[5], Frederick Zittrell[5], Marie Dacke[1], Eric Warrant[7], Gerit Pfuhl[4,8], Jürgen Rybak[9], Kevin Tedore[1]

[1]Department of Biology, Lund University, Lund, Sweden; [2]NanoLund, Lund University, Lund, Sweden; [3]Biocenter, Behavioral Physiology and Sociobiology, University of Würzburg, Würzburg, Germany; [4]Department of Psychology, Chemosensory lab, Norwegian University of Science and Technology, Trondheim, Norway; [5]Fachbereich Biologie, Tierphysiologie, and Center for Mind Brain and Behavior (CMBB), University of Marburg and Justus Liebig University Giessen, Marburg, Germany; [6]Institut für Biologie - Neurobiologie, Free University, Berlin, Germany; [7]Research School of Biology, Australian National University, Canberra, Australia; [8]Department of Psychology, UiT The Arctic University of Norway, Tromso, Norway; [9]Department of Evolutionary Neuroethology, Max Planck Institute for Chemical Ecology, Jena, Germany

**Abstract** Insect neuroscience generates vast amounts of highly diverse data, of which only a small fraction are findable, accessible and reusable. To promote an open data culture, we have therefore developed the InsectBrainDatabase (*IBdb*), a free online platform for insect neuroanatomical and functional data. The *IBdb* facilitates biological insight by enabling effective cross-species comparisons, by linking neural structure with function, and by serving as general information hub for insect neuroscience. The *IBdb* allows users to not only effectively locate and visualize data, but to make them widely available for easy, automated reuse via an application programming interface. A unique private mode of the database expands the *IBdb* functionality beyond public data deposition, additionally providing the means for managing, visualizing, and sharing of unpublished data. This dual function creates an incentive for data contribution early in data management workflows and eliminates the additional effort normally associated with publicly depositing research data.

*For correspondence: stanley.heinze@biol.lu.se

## Introduction

Data are the essence of what science delivers - to society, to researchers, to engineers, to entrepreneurs. These data enable progress, as they provide the basis on which new experiments are designed, new machines are developed, and from which new ideas emerge. Independent of the research field, many terabytes of data are produced every year, yet only a small fraction of these data become openly available to other researchers, with even less penetrating the invisible wall between the scientific community and the public (*Mayernik, 2017*). While research papers report conclusions that are based on data and present summaries and analyses, the underlying data most often remain unavailable, despite their value beyond the original context. Whereas this is changing in many fields and the use of open data repositories becomes increasingly mandatory upon publication of a research paper, this is not ubiquitous and older data remain inaccessible in most instances.

**eLife digest** Insect neuroscience, like any field in the natural sciences, generates vast amounts of data. Currently, only a fraction are publicly available, and even less are reusable. This is because insect neuroscience data come in many formats and from many species. Some experiments focus on what insect brains look like (morphology), while others focus on how insect brains work (function). Some data come in the form of high-speed video, while other data contain voltage traces from individual neurons. Sharing is not as simple as uploading the raw files to the internet.

To get a clear picture of how insect brains work, researchers need a way to cross-reference and connect different experiments. But, as it stands, there is no dedicated place for insect neuroscientists to share and explore such a diverse body of work. The community needs an open data repository that can link different types of data across many species, and can evolve as more data become available. Above all, this repository needs to be easy for researchers to use.

To meet these specifications, Heinze et al. developed the Insect Brain Database. The database organizes data into three categories: species, brain structures, and neuron types. Within these categories, each entry has its own profile page. These pages bring different experiments together under one heading, allowing researchers to combine and compare data of different types. As researchers add more experiments, the profile pages will grow and evolve. To make the data easy to navigate, Heinze et al. developed a visual search tool. A combination of 2D and 3D images allow users to explore the data by anatomical location, without the need for expert knowledge. Researchers also have the option to upload their work in private mode, allowing them to securely share unpublished data.

The Insect Brain Database brings data together in a way that is accessible not only to researchers, but also to students, and non-scientists. It will help researchers to find related work, to reuse existing data, and to build an open data culture. This has the potential to drive new discoveries combining research across the whole of the insect neuroscience field.

Additionally, merely meeting the data deposition requirement by 'dumping' poorly annotated raw files on an internet platform does not aid transparency or reuse of the data. To ensure common standards for data repositories and the datasets to be stored in them, the FAIR principles for data deposition (Findability, Accessibility, Interoperability, and Reusability) were developed (*Wilkinson et al., 2016*). It is clear from these principles that annotation and rich metadata are essential, if a dataset is supposed to be beneficial to others. While this is relatively easily achievable for data such as gene sequences, protein sequences, or numerical datasets, the challenges are much bigger for complex morphological data, physiological observations, or behavioral studies. The difficulties result not only from large file sizes of image stacks, high-speed videos, or recorded voltage traces, but also from the heterogeneous data structure often generated by custom designed software or equipment.

Insect neuroscience is no stranger to these challenges. Particularly for research outside the genetically accessible fruit fly *Drosophila*, no universal data repository exists that allows retrieval of original observations that underlie published articles. Research groups worldwide investigate the nervous systems of a wide range of insect species, but mostly operate in isolation of each other. Data from these projects are often deposited in local backup facilities of individual institutions and thus remain inaccessible to the community. Combined with a lack of interoperability caused by independent choices of data formats this has the potential to severely hamper progress, given that interspecies comparison is one of the existential pillars on which insect neuroscience rests. The problem is amplified by the fact that much of the data are both large and heterogeneous (e.g. 2D and 3D images, models of brain regions, digital neuron reconstructions, immunostaining patterns, electrophysiological recordings, functional imaging data).

A final problem is that depositing well-annotated data takes time and effort, and little incentive is generally given to prioritize this work over acquiring new data or publishing research papers. While this is true for all research fields, the complex data in neuroscience requires an extra amount of effort to meet acceptable standards. This has made depositing data in a form that is useful to the community a relatively rare event. Early efforts were made to develop brain databases for various insect

species (e.g. honeybee [*Brandt et al., 2005*], *Manduca sexta* [*El Jundi et al., 2009a*], *Tribolium castaneum* [*Dreyer et al., 2010*], desert locust [*Kurylas et al., 2008*]), but in those cases, the anticipated interactive platforms for exchange and deposition of anatomical data were short-lived and not used beyond the laboratories that hosted them. More recently, several successful databases for anatomical data from insects were developed. Most notably, Virtual Fly Brain (VFB) (*Osumi-Sutherland et al., 2014*) now bundles most efforts in the *Drosophila* community regarding the deposition of neuroanatomical data - including single-cell morphologies from light microscopy, data from recent connectomics projects (most notably from *Scheffer et al., 2020*), as well as catalogues of GAL4 driver lines, which enable access to specific neurons with genetic methods. VFB hosts most content of older independent databases, such as FlyCircuit (*Chiang et al., 2011*) and FlyBrain (*Armstrong et al., 1995*) and is dedicated to providing smart ways of utilizing and visualizing *Drosophila* neuroanatomy. Similarly, but more focused on data visualization and connectivity modeling, the FruitFlyBrainObservatory allows access to current datasets of single neuron morphologies from *Drosophila*. Another database, founded in 2007, has grown substantially over recent years: NeuroMorpho.Org (*Ascoli et al., 2007*). It provides 3D reconstructed datasets of more than 100,000 neurons from across animal species and includes substantial numbers of single-cell data from insects. While the latter platform is comparative in nature, it does not offer dedicated tools for comparisons between species or much context for the deposited neuron skeletons. In contrast, the data on VFB are much richer and different datasets are tightly linked to each other, allowing, for example, correlation between GAL4 driver lines and electron microscopy based single neuron reconstructions. Yet, no comparison to other species is possible or intended via VFB. Additionally, no systematic information is provided about the function of the deposited neurons in either database, precluding insights into the structure-function relations that are critically important to understanding the insect brain.

To address these shortcomings we have developed the Insect Brain Database (*IBdb*), a cross-species, web-based platform designed for depositing and sharing research data that consist of morphological and functional information from insect brains. With an overall modular design, a concept for dual use as depository and data management tool, combined with widely useful visualization tools, this database yields a tool for increasing transparency, accessibility, and interoperability of insect neuroscience data. Moreover, the newly developed concepts are not only relevant to insect neuroscience, but to any scientific field that can be linked to a hierarchically organized framework. We thus hope that our conceptual design can be adopted by a range of users from across the sciences to simplify data handling and make scientific results in general more transparent.

## Results

### Database outline

The 'Insect Brain Database' (*IBdb*) can be found on the internet at insectbraindb.org and is freely available to everyone. It can be used with most modern web browsers with active Javascript (tested with Google Chrome, Safari, Firefox) on computers running any operating system, without the need for any additional plugins. A user account can be registered free of charge and is required for users who wish to download data and to contribute content.

The *IBdb* is divided into three main hierarchical layers: Species, brain structures, and neuron types. Each level is additionally linked to 'experiments', which is the fourth major organizational layer of the database (*Figure 1A*). At each level, a database entry is represented by profile pages, on which all relevant information is collected (*Figure 1C*). These profile pages are the core of the database. They can be reached directly by several search functions, as well as by entry lists. As profile pages are embedded in a hierarchical framework of species, brain regions, and cell types, they become automatically associated with metadata. For instance, an entry for a pontine neuron of the fan-shaped body (central body upper division) in the monarch butterfly would become linked to the brain regions 'fan-shaped body/central body upper division' and its parent region 'central complex', as well as to 'monarch butterfly' as species. The neuron can then be found, for example, by querying the brain regions associated with it, or by exploring the species of interest.

Species entries contain representative data of an insect species, with the aim of defining that species and the overall layout of its brain. Similarly, entries for brain regions and cell types contain representative examples of data that illustrate the respective entity and provide all information to

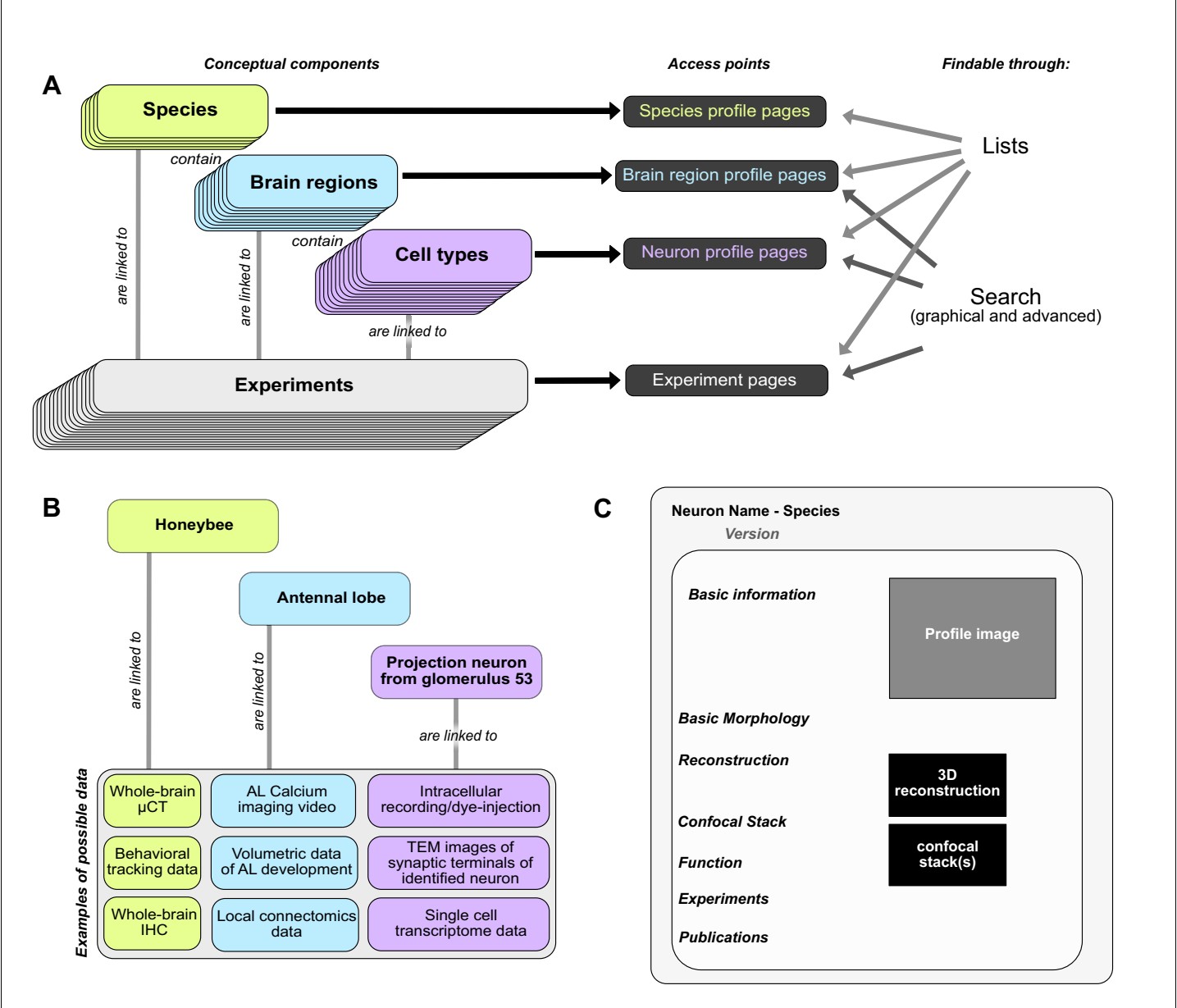

**Figure 1.** Basic concepts behind the Insect Brain Database. (**A**) Organizational layers of the database. Elements in each layer are represented on their respective profile pages, which can be located either through lists or search interfaces. (**B**) Examples for information that can be deposited on each level of the database, illustrating how diverse data is automatically associated with hierarchical metadata. (**C**) Schematic illustration of a neuron profile page. Detailed anatomical and functional data is available on this page, from where experiments associated with this cell type are also linked. Similar pages exist for species and brain regions. Abbreviations: AL, antennal lobe; IHC, immunohistochemistry; TEM, transmission electron microscopy; μCT, micro computed tomography.

The online version of this article includes the following figure supplement(s) for figure 1:

**Figure supplement 1.** Technical overview of database layout.
**Figure supplement 2.** Definition of cell types.

unambiguously define it, essentially providing type-specimen. While the definition of species and brain regions are straight forward in the context of this database, neuron types can theoretically be defined in many different ways, based on connectivity, function, developmental origin, morphology or neurochemical identity. Given that the database is predominantly organized according to anatomical principles, we have defined a cell type as a collection of individual neurons that are morphologically indistinguishable at the level of light microscopy. In many cases, a cell type comprises only a

single individual per brain hemisphere, but in cases where multiple identical neurons are present, a cell type is defined as the first level of similarity beyond the individual neuron. Higher levels of neuron categories group neurons according to anatomical similarities (*Figure 1—figure supplement 2*).

Contrary to the first three levels, experiment entries contain specific data from individual, defined experiments (*Figure 1B*). As research data can be obtained at the levels of species, brain regions and cell types, experiment entries exist for all three levels and are accessible via the respective profile pages (*Figure 1A*). This distinction between representative and concrete data is important, as for example only one entry for a certain columnar neuron type of the central complex exists in the database, yet, if that single cell type was subject to 30 intracellular recordings, the profile page of that cell type would list 30 experiments, each containing a unique individual neuron with its associated physiology. While an anatomical type specimen can easily be defined as the most complete example of any particular neuron's morphology, the decision of what content to depict as representative functional data is less straight forward. The functional information on the profile page should reflect the range of experimental results present in the experiment entries of any particular cell type. As those results potentially diverge, for example due to different experimental paradigms used in different research groups, the function section can contain as many entries as needed to capture the available information, without need for consensus. With new experiment entries added, this section can evolve over time.

In the context of this evolving content, it is key to ensure that entries which are cited in published work remain findable in the exact form they existed when they were cited (*Ito, 2010*). Therefore each entry receives a persistent identifier. This identifier (a 'handle') links to a version of the entry that was frozen at the time it was created (and cited). Once information is added or removed from that entry, a new handle must be generated, providing a new access link for future citations. This system is applied to multiple levels of the database (experiments, neurons, and species) and ensures that all information in the database, as well as the interrelations between entries, are truly persistent.

## Interactive search interfaces

As locating specific datasets is one of the core functions of a database, we have developed a novel, more intuitive way to find specific neuron data. A graphical representation of the insect brain, resembling the overall anatomical outline of all brain regions and highlighting the regions containing neuron data (*Figure 2A*), makes it easy to search for neurons within single species and across species. This graphical interface is generated directly on the database website for each species and is adapted from a generic insect brain, that is a shared ground plan. This generic brain is the least detailed fall-back option for any cross-species search and was generated based on the insect brain nomenclature developed by *Insect Brain Name Working Group et al., 2014*. It resembles the shared anatomical hierarchy of all brain regions in insect brains. Within this hierarchy, the entire brain is divided into 13 super-regions, which consist of individual neuropils. The latter can be further divided into sub-regions. While all super-regions exist in all species, differences become more pronounced at lower levels of the hierarchy. The generic brain therefore largely contains super-regions as well as several highly conserved neuropils (*Figure 2—figure supplement 1*, *Figure 2—figure supplement 2*). As these categories are simply tags of brain region entries used to organize the database, the search interface does not differentiate between them, simplifying the user experience (*Figure 2A*).

If more than one species is subject to a query, a schematic brain is generated that displays the commonly shared features of the species involved. For both single and multi-species search, when selecting a specific brain region, all neurons in the *IBdb* that connect to this region become visualized by a dynamically drawn wiring diagram (*Figure 2B*). Filters can be applied to narrow down search results according to neuron polarity, functional class, etc. Individual neurons in the wiring diagram can be selected to reveal the neuron's profile page. Here, all available information for this cell type is displayed, including links to deposited experiment entries (*Figure 2C*). The schematic display of search results can visualize any neuron in the database, only requiring that a neuron is annotated with respect to the brain regions it innervates. A list of search results is additionally made available after each search and offers the possibility to also display experiment entries associated with the found neurons.

For single species queries, two more modes for visualizing search results are available: the semi-schematic view and the 3D view. The semi-schematic view mode emphasizes the natural brain

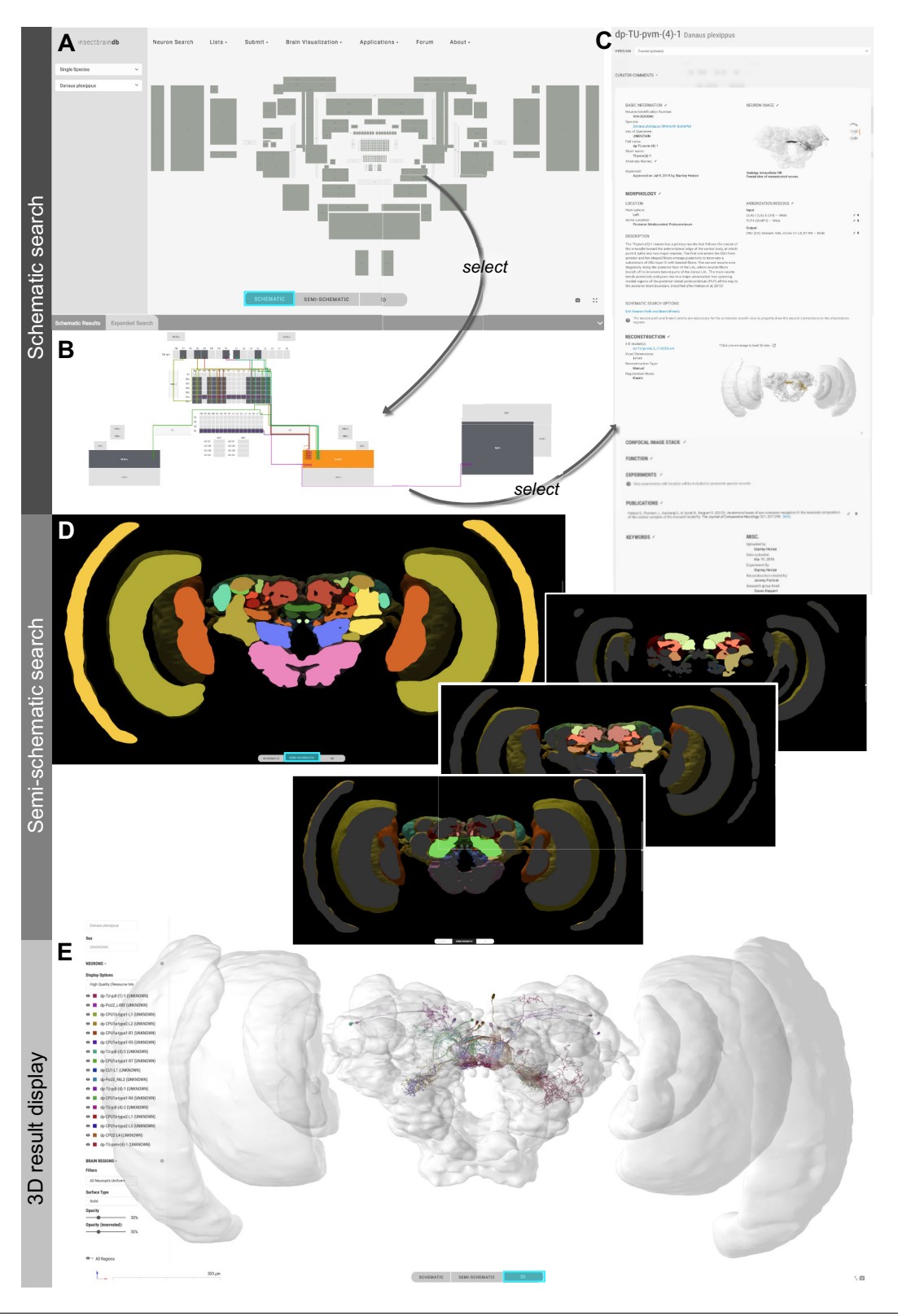

**Figure 2.** Neuron search in the *IBdb*. (**A**) Screenshot of the schematic brain search interface (Monarch butterfly) in single species mode. Selecting a neuropil will reveal all neurons connected to that neuropil. (**B**) Schematic wiring diagram view of the search result; orange neuropil was queried. Selecting an individual neuron will reveal the profile page of that cell type. (**C**) Example of a neuron profile page with anatomical information. Confocal image stack and functional data are not listed in this example. (**D**) Semi-schematic search interface. The section view is scrollable and allows the user to

*Figure 2 continued on next page*

*Figure 2 continued*

query individual neuropils for connected neurons by clicking the cross section. The inserts show the results view at three levels of the brain. Neuropils connected to the queried neuropil are highlighted. Switching to the schematic view will then show the neurons as wiring diagram. Switching to the 3D mode will show registered neurons in 3D. (E) The 3D results viewer allows one to view all neurons registered into a common reference frame; example from Monarch butterfly (data from *Heinze and Reppert, 2012*) . The user can continuously switch between the three modes (schematic, semi-schematic, 3D). All visualizations generated with Insectbraindb.org.

The online version of this article includes the following figure supplement(s) for figure 2:

**Figure supplement 1.** Schematic outline of generic brain and variations across species.
**Figure supplement 2.** Brain structures in the database and their hierarchical organization.

organization on the level of brain regions, while also serving as interface for launching search queries. It comprises a full series of automatically generated sections through a segmented 3D brain of a species. Each brain region present in that species' 3D brain is shown as an interactive cross section that can be used to query neural connections of that region (*Figure 2D*). If a region is selected, all connected brain areas are highlighted and neurons resulting from this query can be visualized by changing to either the schematic view, the 3D view, or a list view. The advantage of the anatomically correct layout of this interface is that a brain region can be queried for a neuron, even if its name is not known to the researcher. This is particularly useful for regions with uncommon names that have only recently been introduced to the insect brain naming scheme (e.g. crepine, superior clamp, etc., see *Insect Brain Name Working Group et al., 2014*). Launching a search for neurons in regions with unfamiliar names is made much easier when the search interface resembles the information a researcher has obtained from, for example, confocal images or physical brain slices. The semi-schematic mode of the database search function fulfills this demand and bridges the schematic wiring diagram view and the full 3D view.

The 3D view visualizes search results in an anatomically correct way and shows queried neurons in the context of a species' reference brain (given that this information was added) (*Figure 2E*). It displays interactive surface models of that brain together with neuron skeletons obtained from the neuron-type's profile page.

In the graphical search interface the search parameters are limited to anatomical information defining the neuron's location in the brain (i.e. likely input and output areas). In contrast, an additional text-based search function ('Expert Search') allows users to query all information deposited on a neuron's profile page. Individual search parameters can be logically combined to generate arbitrarily complex searches. The results are displayed as a list of neurons, which can be sent to populate a schematic wiring diagram view by a single click. Thus, this tool effectively combines complex search with the advantages of an intuitive display of results.

A different means of locating data on neurons and experiments is achieved via a publication based search function. Each neuron and experiment that is associated with a publication becomes automatically part of a dataset linked to this publication. By definition, experiments are only part of one publication, while neuron entries can be referenced by many publications. In either case, users can locate all data that has contributed to a specific piece of scientific literature.

Finally, whereas the emphasis of the database search lies on locating cell types, information on brain regions can also be found using identical interfaces. The schematic search option allows users to reveal brain region profile pages by selecting schematic neuropil representations. The same information can also be obtained by selecting brain regions in the semi-schematic neuropil search interface.

## Online applications and tools

To maximize the usefulness of the database, we have implemented an integrated 3D viewer to deliver platform independent, high-quality data visualization without any additional software demands. Neuropil visibility can be independently switched on and off for each brain region, transparency can be freely adjusted, and colors of neurons can be changed (*Figure 3A*). Neurons can be shown either with diameter information or as simple backbones. The built-in screenshot function enables the user to capture any scene displayed in the 3D viewer and produces a high-resolution, publication-ready image with transparent background (*Figure 3B,C*).

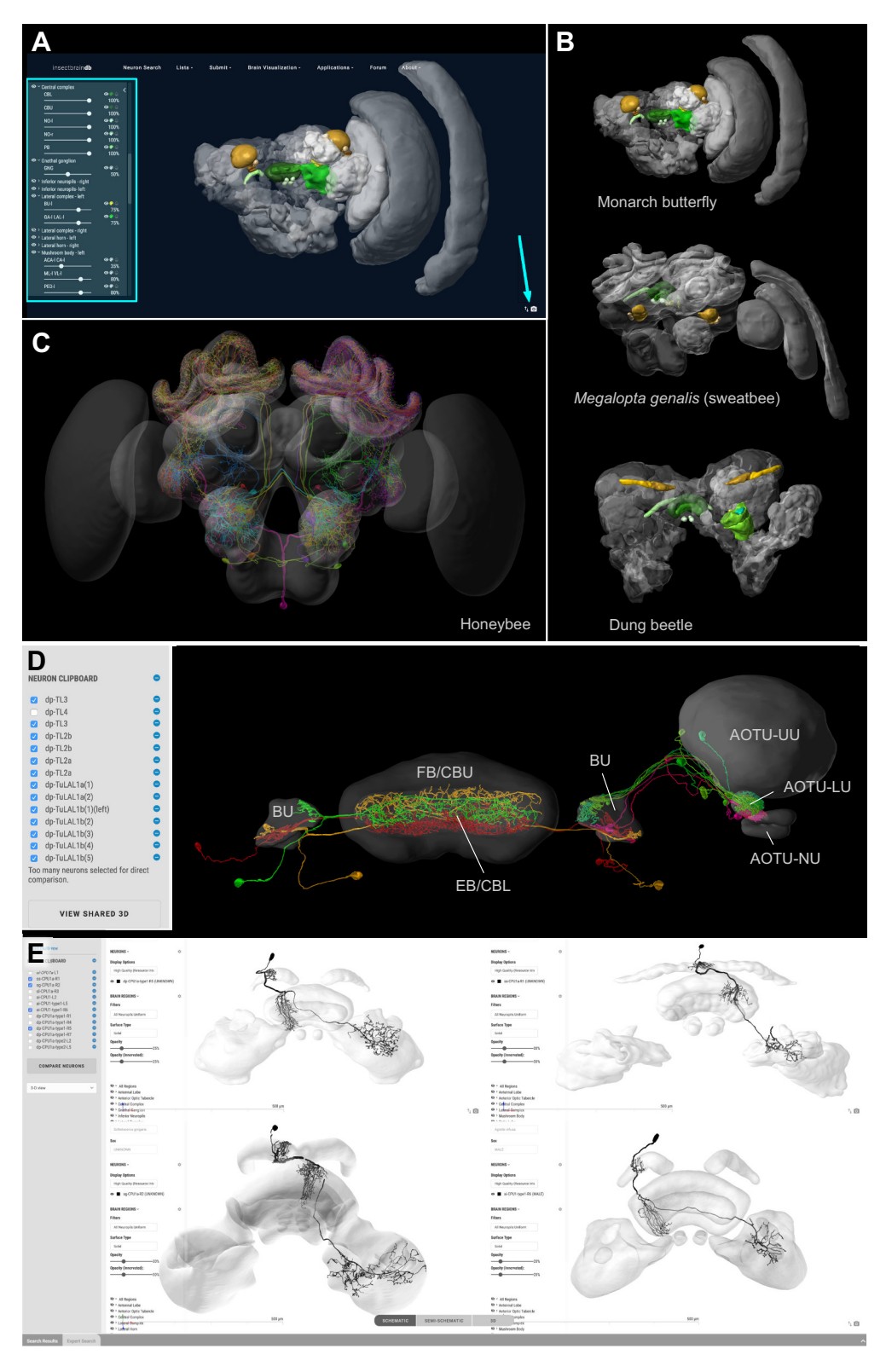

**Figure 3.** Visualization tools and applications. (**A**) Screenshot illustrating the functionality of the 3D viewer in the insect brain database. Cyan arrow: Screenshot button. Cyan panel: Tools for adjusting appearance of neuropils. (**B**) Examples of neuropil images generated with the *IBdb* 3D viewer, illustrating navigation relevant neuropils in three insect species (Monarch butterfly [***Heinze and Reppert, 2012***], sweat bee *Megalopta genalis* [***Stone et al., 2017***], dung beetle [***Immonen et al., 2017***]). (**C**) Neurons associated with the antennal lobe of the honeybee, generated with the *IBdb*

*Figure 3 continued on next page*

*Figure 3 continued*

3D neuron viewer (data from ***Rybak, 2012***). (D) Elements in the neuron clipboard (left) can be arbitrarily combined and displayed in the 3D viewer to highlight neural pathways and circuits. Shown are two parallel input pathways from the anterior optic tubercle to the ellipsoid body of the central complex in the Monarch butterfly (data from ***Heinze et al., 2013***). (E) Side-by-side neuron viewer. Screenshot showing comparison of 3D skeletons of CPU1 (PFL) neurons from four species (top left: Monarch butterfly [***Heinze et al., 2013***]; bottom left: desert locust [***El Jundi et al., 2009b***]; top right: Dung beetle [***el Jundi et al., 2015***]; bottom right: Bogong moth [***de Vries et al., 2017***]). All visualizations generated with Insectbraindb.org.

The *IBdb* allows users to not only locate neuronal morphologies quickly, but also to combine arbitrary neurons from any single species into a common visualization. To achieve this, we have generated a neuron clipboard, in which individual neurons from search results can be stored temporarily (***Figure 3D***). Any subset of cells in the clipboard can be sent to the 3D viewer, as long as all neurons belong to the same species, that is can be displayed using the same reference brain. The desired configuration of neurons and neuropils can be generated using the interactive tools of the viewer and the screenshot function can be used to create a high-resolution image to be used for illustration purposes (e.g. reviews, conference talks, teaching).

Additionally, we have embedded a function to directly compare up to four neurons side by side on screen. Any neuron located in the neuron clipboard can be chosen to be included in this comparison. The comparison uses the 3D view, the profile image, or the confocal stack located on the respective neurons' profile pages. This function is ideally suited to compare homologous neurons from across species to quickly assess differences and shared features of these cells. The four-window 3D viewer retains all functions of the normal full screen 3D viewer and thus also allows the capture of high resolution screen shots of each of the neurons being compared (***Figure 3E***).

The data in the database are suited for many applications, including more sophisticated ones. To provide direct access to all levels of the data in the *IBdb* we have created an API interface, specifying how to automatically draw data from the database via web-browser based apps. Applications produced by third parties that use this function can be embedded directly on the *IBdb* website, once they are approved by the site administrators. Applications envisioned are, for example, quantitative comparisons of both single neuron morphologies and neuropils between species, direct online multi-compartment modeling of neurons deposited in the database, or virtual reality interfaces that allow exploration of anatomical data in a 3D virtual reality environment. Over time, we hope that our unified platform will stimulate the insect neuroscience community to generate a collection of online tools to analyze and explore neuroanatomical and physiological data deposited in the *IBdb*, thereby allowing straight-forward meta-analysis of all raw data deposited in the database. As an offline tool, the Natverse package in R already offers the possibility to explore *IBdb* data (***Bates et al., 2020***).

## Contribution of data

All data on the internet is public. This also applies to any data publicly available in the *IBdb*. Driven by the requirement to obtain data persistency and implemented by the use of handles, no data can be removed from the database once it is public (and thus citable). For all data, the contributors retain ownership and hold the copyright to their data. The publishing is performed explicitly by the owners, not by the database administrators, and the license attached to each dataset is a Creative Commons Attribution Non Commercial 4.0 International (CC BY NC 4.0). Thus, when data in the *IBdb* are downloaded for reuse, the original work that underlies these data has to be credited together with the *IBdb* as the source. When data are used to generate images with the help of the *IBdb*, these images are licensed via Creative Commons Attribution 4.0 International (CC BY 4.0), that is they can be used in any publication as long as the original data owners and the *IBdb* are credited.

Data can be contributed by registered users at all levels of the database, i.e. species, brain regions, neurons, and experiments. The process is similar on all levels but requires more expert knowledge the higher in the hierarchy the data reside. In the following sections we will briefly illustrate the main principles of how to contribute data (for full instructions see the Online User Guide).

### Species

To submit a new species, a profile page has to be created, which then has to be populated with data. While photographs, bibliography, and text descriptions of the species are desirable and

strongly encouraged, the most important next step is to generate a schematic brain (*Figure 4A*). This will ensure that brain structure entries are created in the database, a prerequisite for neuropil-based search and 3D brain region identity. To generate a schematic brain, we have created the 'Brain Builder' tool on the *IBdb* website, which provides templates based on either the generic insect brain, or related species already deposited in the database. The user can simply copy an existing brain, associate it with the new species and modify it to match any unique features of the new species. Note that the ventral nerve cord (VNC), even though not part of the brain, was included as a single region as well, allowing to deposit information about VNC neurons as well and opening the possibility to expand the database beyond the actual brain in the future.

Once the profile page and the schematic brain is created, a 3D brain can be uploaded to illustrate the brain organization of the new species and to serve as reference brain for neuron display (*Figure 4A*). For all uploaded data, the database distinguishes between source files and display files. Source files contain the 3D reconstruction in a format that the researcher would like to make available for others in the field. The second set of files, the display files, are required for automatic online display and must constitute the surface models of each neuropil (.obj-format). Each brain region model is tagged with a unique neuropil identity, so that the schematic brain regions and the 3D surface models will be linked to the identical brain-structure entry. Finally, following the same principles

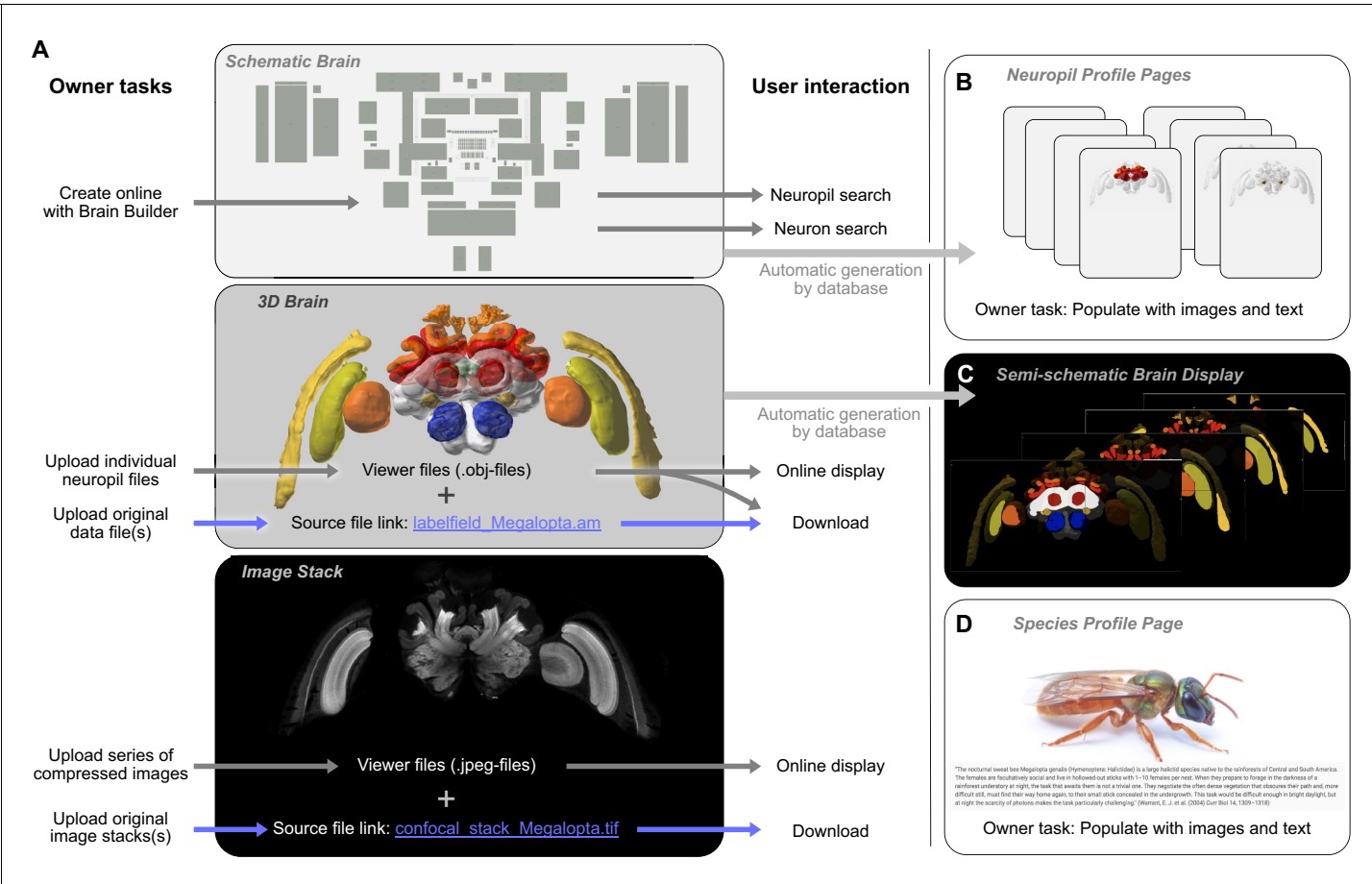

**Figure 4.** Contributing a species to the *IBdb*. (**A**) Three main elements have to be created for each new species: the schematic brain, the 3D brain, and an image stack. The schematic brain is generated directly on the *IBdb* website using the 'Brain Builder', while the other two elements are uploaded. For each, both source files and viewer files are needed. Viewer files are used for online display, while source files can be downloaded by users. (**B**) Neuropil profile pages are automatically generated when creating the schematic brain. They have to be populated with images and texts by the user. (**C**) The semi-schematic brain is automatically generated based on the provided 3D brain. (**D**) The species profile page must be populated with images, texts and a bibliography to provide context for the species. Visualizations in A,B,c generated with Insectbraindb.org, *Megalopta genalis* data from *Stone et al., 2017*.

Photograph in D reproduced with permission from Ajay Narendra.

as for the 3D brain, an image stack can be uploaded to the profile page as well (*Figure 4A*). This can be any representative dataset that illustrates the layout of the species' brain (e.g. confocal stack, μCT image series, serial sections with any other technique).

## Brain structures

Brain structure entries are automatically generated when defining the schematic brain search interface for a new species (*Figure 4B*). These profile pages are automatically populated by a 3D brain in which the relevant region is highlighted, a brain structure tree that reveals the relative location of the respective neuropil in the hierarchy of the species' brain, and with links to neuron entries associated with each brain region. All remaining data have to be manually added. These are mostly descriptive in nature and encompass images, text-based descriptions, and volumetric data.

## Neurons

Contribution of neurons follows a similar procedure as the contribution of new species (*Figure 5*). The user creates a profile page for the new cell type that subsequently has to be populated with information. To make a neuron findable in the database, its arborization regions in the brain have to be defined (*Figure 5C*). Within a graphical user interface, these regions are chosen from the brain

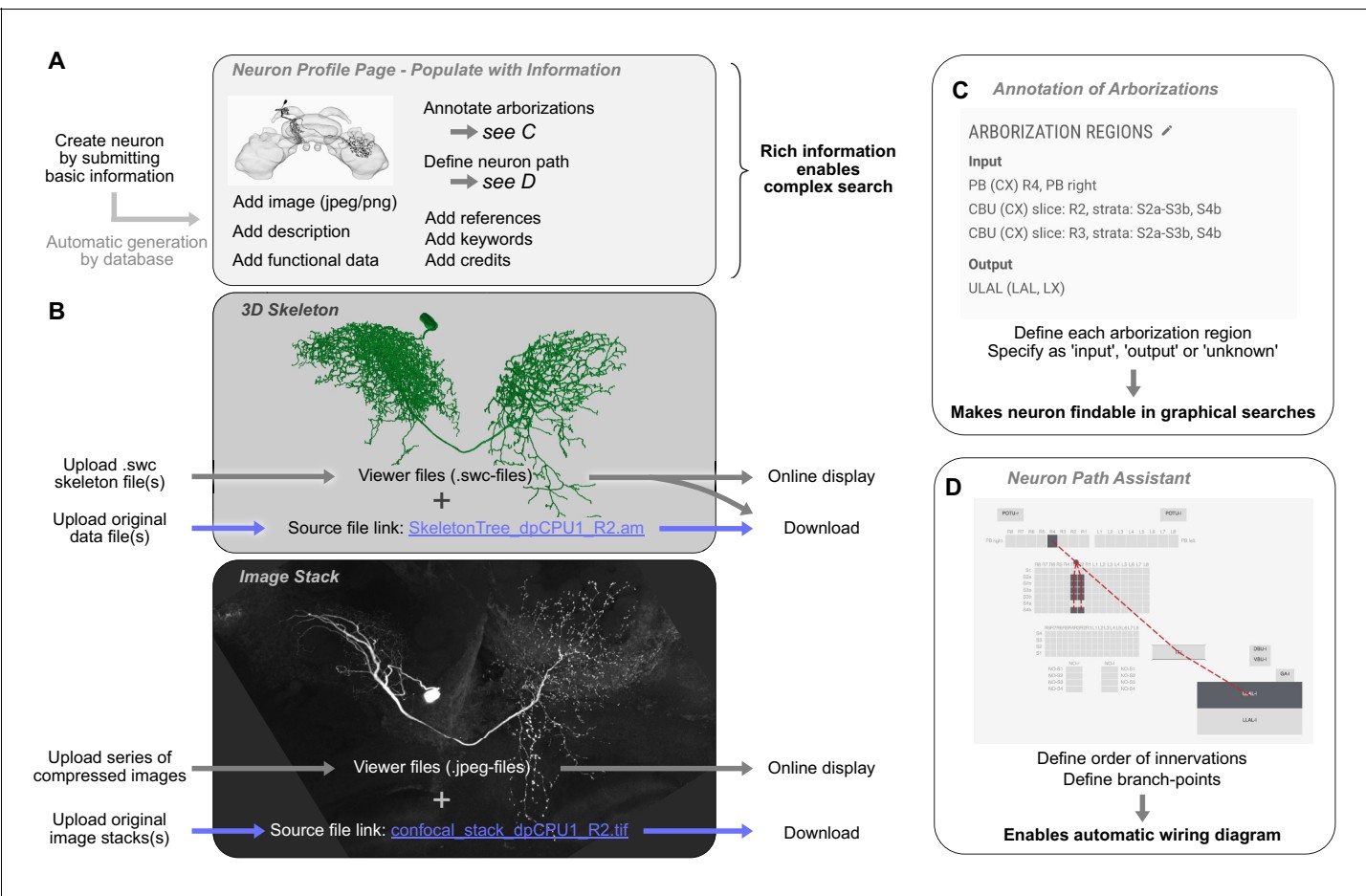

**Figure 5.** Contributing a neuron type to the *IBdb*. (**A**) A new neuron type can be created by submitting a neuron form containing basic information. This generates a neuron profile page that then has to be populated with information by the owner. Each entry is findable by the expanded search function. (**B**) 3D-skeletons are added as swc-files (online display) and source files (download). Confocal image stacks are uploaded as jpeg series for online display and as original data files for download. (**C**) All arborization regions of the neuron must be defined (at the level specified in the species' schematic brain) and labeled as either input, output, or unknown polarity. (**D**) To enable automatic drawing of wiring diagrams in schematic search results, the order of innervation of neuropils and the branch-points of the neuron must be defined using the path assistant. All visualizations generated with Insectbraindb.org.

structures available in the schematic brain of the respective species. One arborization entry has to be created for each branching domain of the neuron, leaving no part of the neuron un-annotated. To enable the automatic generation of a wiring diagram view of the new neuron for displaying schematic search results, an outline of its branching structure has to be generated in an embedded tool called the neuron-path assistant (*Figure 5D*). This branch tree defines which neuropils are innervated in which order and where main branch points are located.

The remaining procedure for neuron contribution is largely identical to species contribution and follows the dual approach towards source data and display data for 3D reconstruction and image stacks (*Figure 5B*). All other information, that is images, bibliography, keywords, representative functional data, transmitter content, and textual descriptions, can be added to the profile page at any time prior to publication. To ensure common minimal standards and to avoid rudimentary datasets, several data fields have to contain valid information before a request for publication can be made. These include an image of the neuron that allows to clearly identify the cell type, a complete morphology section (including soma location, a description of the neuron's morphology, and arborization regions), as well as a link to a publication (or preprint) that contains data on the cell type depicted in the new entry. If no publication is available, a note in that section will state that the neuron is not part of any publication.

Importantly, the *IBdb* can be used to house data that have been obtained by classical methods, for example camera lucida drawings of Golgi impregnated neurons. While no 3D information is available in those cases, drawings can be uploaded as images after which annotation of the neuron's morphology (arborization regions) is performed as described. These neurons will therefore become findable in the schematic search interface and will be added to the publicly available pool of neuronal data. To allow these datasets to exist in the database, 3D reconstructions and image stacks are not mandatory content. Similarly, 3D data that is not registered to a common reference frame can be deposited in the database, with or without individual brain region reconstructions, but a shared display in the context of a reference brain will not be available for these datasets.

## Experiments

Experiment entries are created by adding them directly to the profile page to which they are linked (species, brain structure, or neuron). The automatically generated experiment profile page must then be filled with basic meta-information about the experiment (date, what was done, who did it), after which a series of files can be uploaded. These files can be in any format and are made available for download. This allows users to provide not only the raw data of any experiment, but also, for example, analysis scripts, custom made equipment-control software, and analysis results. Image files can be selected for direct online display on the experiment profile page to allow online examination of the data. Importantly, experiment entries are independently published from neuron entries.

## Curation and administration

The database is managed via a group of voluntary curators, a scientific administrator, and a technical administrator. Importantly, no single person curates all data in the *IBdb*, but each species is managed by a specialized curator, who is an expert for that species. This distributed curation system ensures that no single person is responsible for too many datasets, and that no curator has to evaluate data outside their area of expertise. To additionally reduce the workload for species with many entries, more than one curator can be assigned to any given species. The scientific administrator (the lead author of this publication) oversees the curators, while technical administration is carried out by the technical administrator (last author of this publication). The technical administrator is the only person who has potential access to all data in the database.

The responsibility of the scientific administrator is to approve new species and to train and support the curators for individual species. This training is carried out during a training period during which actions of the curator have to be approved by the scientific administrator before they take effect. Once a new curator is sufficiently trained to carry out all tasks independently, the scientific administrator grants full curator privileges. The responsibility of each curator is to approve new neuron-type datasets and to re-evaluate major updates of these (data re-approval). This process entails checking for formal errors in the submitted data, ensuring that new data do not accidentally duplicate already existing data, and that the quality of the data meets acceptable standards. To ensure

swift correction of any issues, we have implemented a private communication channel between curator and data owner. This was realized through a commenting function that enables the curator to post comments on a neuron page, which are only visible to the data owner. The owner can then directly respond to the comments and any issues raised can be resolved.

While our distributed approach to curation has many advantages, it creates challenges towards ensuring that curation of data across all species and levels is accurate, consistent and complete. We have therefore designed multiple tools and features to facilitate effective and consistent data curation. Besides formalized training and approval for each new curator, we have provided checklists for both data contributors and curators (found in the online help menu) that must be followed to ensure that database entries fulfill predefined standards. Yearly meetings among all curators will ensure that these standards are known and can evolve over time. If mandatory data is not provided by a data contributor, request for approval of the dataset is automatically blocked, preventing rudimentary datasets to enter the public section of the database. Finally, if an entry becomes obsolete, for example when new research data provides conclusive evidence that cell types listed as separate entries belong to the same type, existing entries can be archived. This process preserves the persistent handle of the entry, but removes it from public lists and search results in the *IBdb*. This way, references to these datasets remain resolvable, while, at the same time, obsolete data cannot clutter the database. In the long run, this function provides the means to keep the database clean without violating the principle of data persistence.

The definition of what constitutes minimal standards for a neuron or species entry reflects a scientific consensus among curators. Rather than imposing these standards onto the field, the *IBdb* generates the means to develop a common set of rules by providing the platform for continued discussions among curators as well as to reinforce an evolving consensus.

To enable all users to provide feedback and to discuss topics relevant to other database users, we have added a discussion forum directly to the *IBdb* website. This forum is intended as a means for reporting potential bugs, suggesting new *IBdb* features, or for discussing scientific content (methods for data processing or acquisition, requests for literature, staining protocols etc.).

## The *IBdb* as tool for data management and data deposition

Each database entry has to be explicitly published by the contributor. In the process, it is approved by either the database administrator (species), or the species' curator (neurons). While this procedure was initially intended only as a quality control measure to prevent incomplete or inaccurate data from compromising the database, we have developed it into a unique feature: the *IBdb* private mode. Before a dataset is made public, it is invisible to all other users, curators and the scientific database administrator. The dataset can thus be updated and even deleted. This creates the potential of using the *IBdb* to deposit data while they are being collected or prepared for publication in a research paper, that is, for data management (*Figure 6A*). The API can be used to programmatically integrate the *IBdb* into any individual data management workflow; a community driven MATLAB implementation for automatic deposition of cell types and experiments is available on GitHub (https://github.com/zifredder/IBdb-matlab).

To facilitate the use of the *IBdb* as a data management tool, we have enabled three operational modes of the database site: private, public, and mixed. Any user logged in can thus choose to either access only (own) private data, only public data, or both. The first mode turns the *IBdb* into a data management site for ongoing research, the second mode is the default mode for viewing publicly available data, and the third mode allows the user to compare their own unpublished data with public data.

As efficient data management requires researchers from the same laboratory, as well as collaborators, to have access to relevant unpublished data, each user can grant access to their own private datasets (*Figure 7*). To this end, a user can create a user group and invite other database users to join. Datasets can then be added and made visible or editable to all members of the group. These data can either comprise individual entries at all levels of the database, or collections of entries defined by common features.

Finally, users are often reluctant to make datasets available to the public before they are included in a research paper, yet, these data should be available to editors and anonymous reviewers. We have therefore created the possibility for 'pre-publishing' database entries (*Figure 6A*). This function allows a user to assign a persistent handle to a dataset (e.g. a neuron profile page) without approval

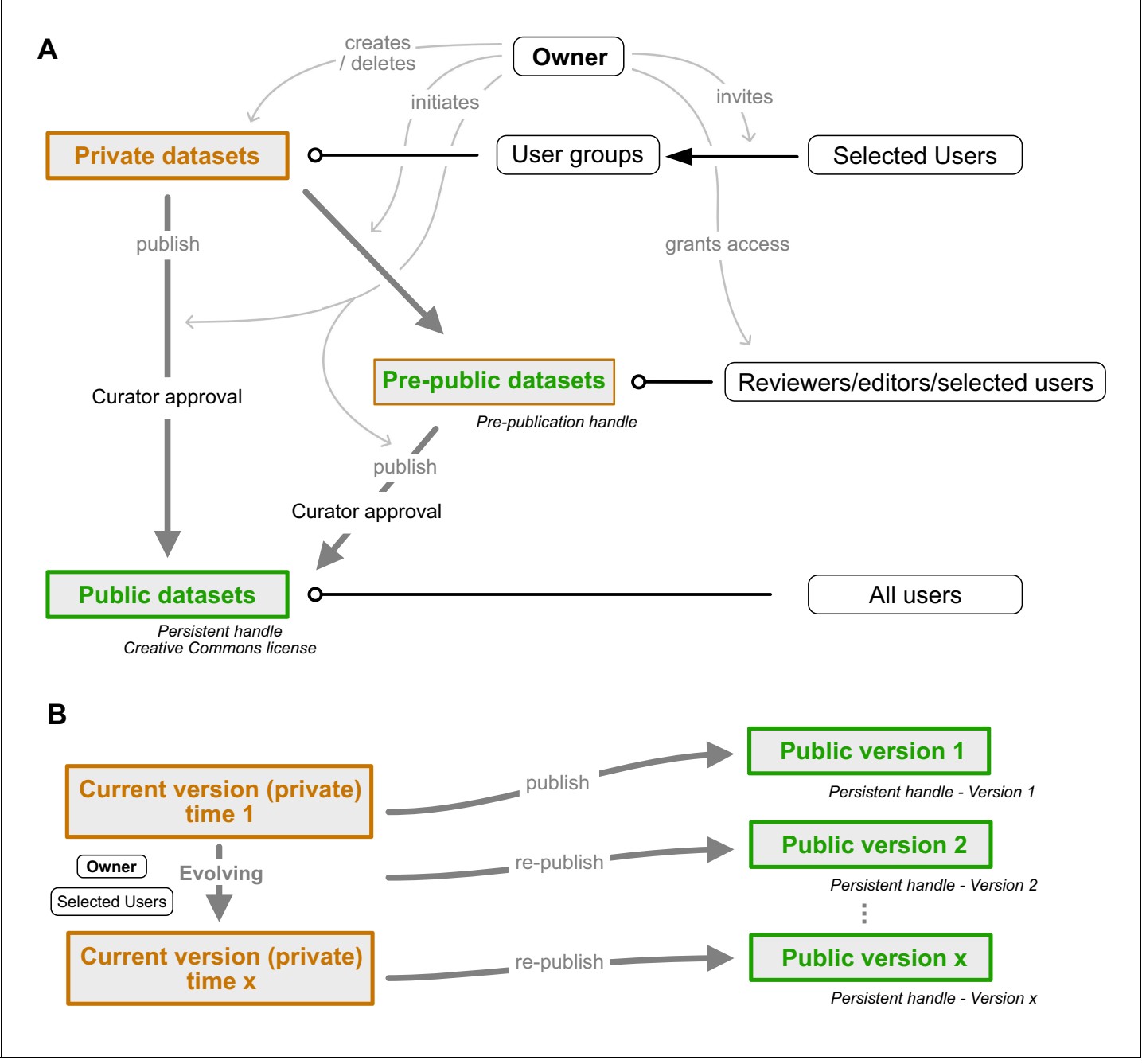

**Figure 6.** Dataset publication concept. (**A**) Interconnection of private, public, and pre-public datasets. Private datasets can be viewed and edited by members of user groups with access granted to a particular dataset. Pre-public datasets can be viewed by anyone in possession of the pre-publication handle (distributed by the data owner, e.g. within a manuscript). Public datasets can be accessed by all users. Publication of data cannot be undone as persistent handles are generated. Gray arrows indicate control actions employed by the dataset owner. (**B**) Re-publication strategy of evolving datasets. A current version of each public dataset remains present in the owner's private database mode and can be edited at wish. Once sufficient updates have accumulated, the dataset can be re-published. A new version of the persistent handle is assigned and the now public dataset (version 2) becomes locked. Datasets can be edited by the owner and anyone who has been granted permission to edit by the owner.

by the curator and without making the dataset findable through search or lists in the *IBdb*. Editors and reviewers of the research paper (anyone in possession of the handle) then have direct access to the linked pages. Once the manuscript is accepted for publication, the user can submit the respective datasets for publication, obtain curator approval, and thus make them findable in the *IBdb* public mode.

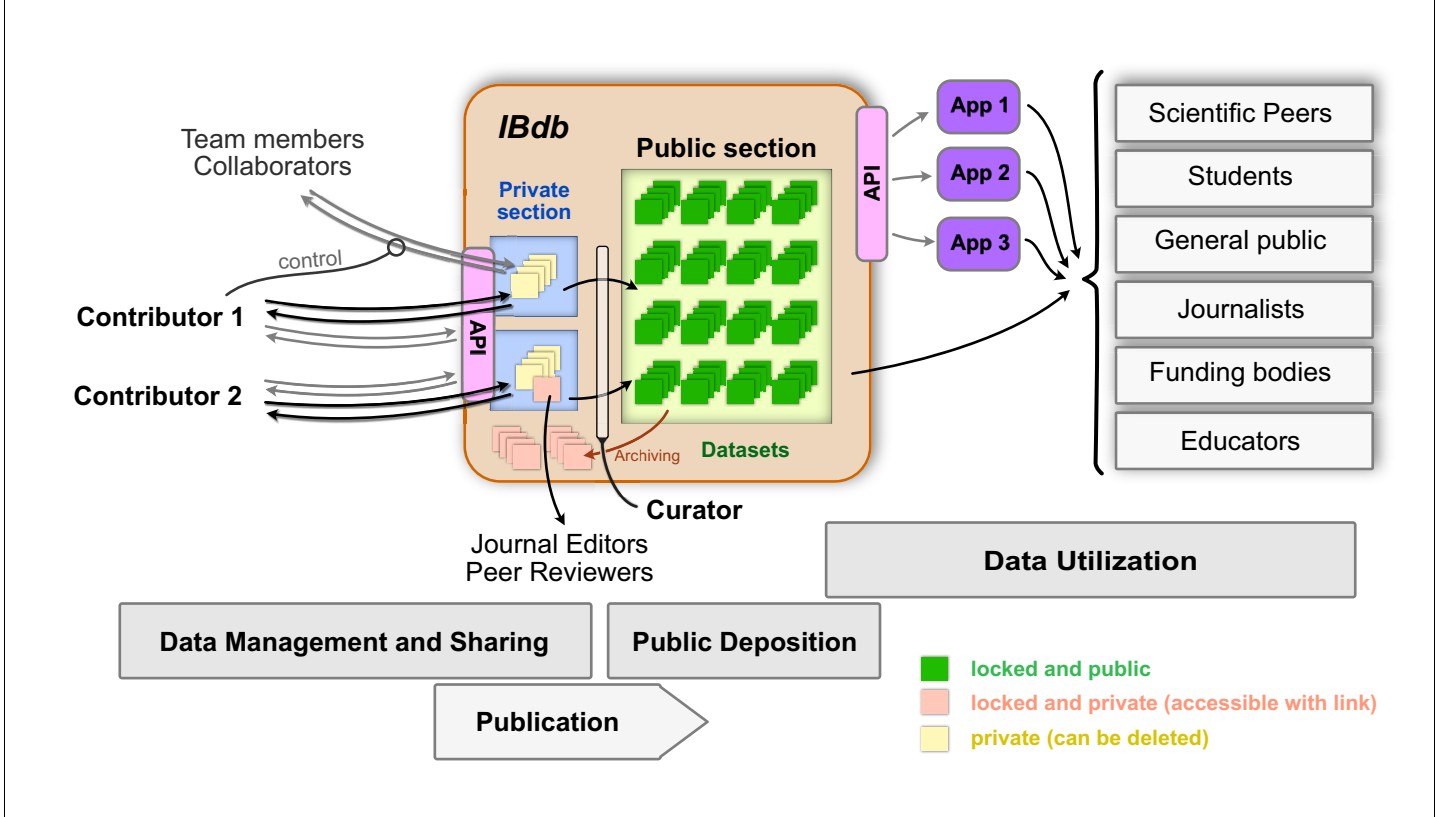

**Figure 7.** The Insect Brain Database and the possible interactions between users and deposited data. The private sections of the database are accessible to only the owner of the data, and datasets within this section can be shared with team members and collaborators. Up- and download of these data are possible either directly or via an application programming interface (API). As these datasets are unlocked, they can continuously be updated and also be deleted. Upon publication and curator approval, datasets become locked (persistent) and are deposited in the public section of the database. As an intermediate step, datasets can be pre-published (locked but private) and made available to journal editors and peer reviewers when including datasets in manuscripts of journal articles. Data in the public section of the database are accessible directly for all interested users (relevant user groups are shown on the right). Additionally, an API also allows automated access of public data, which can therefore be used by third party applications (illustrated as 'App 1–3') for generating specific user experiences with additional capabilities, for instance in the context of teaching. To remove obsolete datasets from the public domain, they can be archived, a process that preserves persistence but prevents datasets from being findable without the explicit handle.

The online version of this article includes the following figure supplement(s) for figure 7:

**Figure supplement 1.** Database usage statistics.

To avoid having to separately provide numerous independent handles when sharing data, individual entries can be grouped into datasets. These receive a unique link that grants access to the entire collection. Only public or pre-public data can be included in datasets.

Public entries of the *IBdb* are maintained in a dual way; the persistent version is locked and cannot be changed, whereas a second, current version remains visible to the owner and to all members of user groups with appropriate access rights (*Figure 6B*). This current version is fully private and can be freely edited or expanded. Importantly, no data that is already part of a public version can be deleted. Rather, when for instance a confocal image stack should be replaced by a better one, the old stack can be archived, so that it will not be visible in new versions of the dataset, but will remain present in the database for display of earlier versions. Once all required updates of a dataset have been made, the edited version can be re-published and will be assigned a new version of the persistent identifier after re-approval by the assigned curator. This new version is now also locked and any further edits will again have to be done in the current version of the dataset. This ensures that all data that have been assigned a persistent identifier will remain valid and accessible, while at the same time allowing each entry to evolve. The described strategy of publishing and re-publishing

and the associated duality of persistent and current versions are implemented at the levels of species, neurons and experiments.

### *Drosophila* and interoperability with Virtual Fly Brain

The *IBdb* does intentionally not include *Drosophila melanogaster* as a species. This is because huge efforts have been spent developing highly efficient resources for this widely used model system and, as a result, the Virtual Fly Brain (VFB) resource has been created (*Osumi-Sutherland et al., 2014*). It bundles data from several older *Drosophila* databases (e.g. FlyCircuit) to the most recent connectomics datasets (*Scheffer et al., 2020*). Serving as the main repository for anatomical data from the *Drosophila* brain it has become the main site to locate GAL4 driver lines, single-cell morphologies, and synaptic connectivity data. It contains tens of thousands of datasets and is designed to specifically meet the needs of the *Drosophila* research community. By being less specialized, the *IBdb* has a wider scope. We are hosting many species and include both functional and anatomical data. We also do not require neuronal anatomies to be registered to a reference brain, if this is not possible for some reason. This opens the *IBdb* up to more diverse data, but as a result cannot provide most of the specialized services that VFB can deliver (e.g. automatic bridging registrations of 3D data between different reference brains). Cross-linking the two databases to effectively enable comparing neurons from the insect species deposited in the *IBdb* with *Drosophila*, nevertheless, was highly desirable. Importantly, both databases have converged on highly similar hierarchical frameworks. As the brain nomenclature used by the *IBdb* and VFB is identical, neuropil identities were mapped across both databases. In cases were homology between corresponding neuropils is unclear, regions map to the next higher order brain region (e.g. Monarch butterfly dorsal bulb maps to the entire *Drosophila* bulb). For some neuropils, there are no known counterparts in the fly (e.g. the posterior optic tubercle).

We use the APIs of both databases to allow direct communication between the *IBdb* and VFB. In principle, when searching the *IBdb* for neurons, an API mediated query can be automatically sent to VFB and search results are displayed as a list of single neuron entries. Each item on the list contains information obtained from VFB and is directly linked to the corresponding entry at VFB. This feature is available for the graphic, brain region based search. This includes complex multi-neuropil searches, aimed at identifying neurons connecting several brain regions. Effectively, this enables a user to launch a query in the *IBdb* and directly view corresponding neuron data in VFB, including data from recent connectomics efforts (*Scheffer et al., 2020*). This seamless interoperability makes maximal use of both complementary resources, without duplicating functionality.

## Discussion

### Specific problems solved by the *IBdb*

Previous and current online databases hosting insect neuroscience data have suffered from several shortcomings: Most severely for old databases, a lack of maintenance often quickly led to outdated file formats, rendering the deposited information no longer compatible for viewing with current web browsers (anticipated by *Ito, 2010*). Second, while some databases originally allowed interactive viewing of the data, no possibility for contribution of one's own data existed and data download was limited to very few files, for example *Brandt et al., 2005*, *Kurylas et al., 2008*. Usability of larger databases was generally impaired by a layout that often required expert knowledge to be able to launch meaningful database queries or to understand search results (e.g. FlyCircuit, Invertebrate Brain Platform [now: Comparative Neuroscience Platform]). This not only applies to old databases, but the restriction to individual species and often highly complex interfaces limit the potential user base to specialists even in cutting edge databases such as VFB or visualization tools such as FruitFlyBrainObservatory. Finally, the limitation to purely anatomical data, including in the major current cross-species database NeuroMorpho.org, does not account for one of the key advantages of insect neuroscience: the high level of tractable structure-function relations.

The *IBdb* addresses each of these issues. Firstly, we have developed the database software to be independent of the operating system and type of web browser used, as well as to not rely on any third party plugins. Additionally, we implemented the database as a classic, relational database without experimental data structures (e.g. intelligent, adaptive search), aiming at maximal robustness.

Having created a conceptually simple software that uses standard web-technology with standard file-formats makes continued compatibility and technical maintenance comparably simple.

Second, we have invested substantial effort in making the *IBdb* intuitive to use and visually attractive to provide a positive user experience. The latter factor should not be underestimated in its importance. One of the problems encountered in previous databases were user interfaces that were difficult to use, creating an immediate negative experience when attempting to use a site for the first time and therefore reducing the motivation to interact with it. As for commercial software, we aspired to generate a user interface that is largely self-explanatory, provides immediate visual feedback when a user action was successful, and which clearly shows what actions can be performed. Several years of beta-testing by a multitude of users have streamlined the site to a point at which interacting with the *IBdb* is both straight forward and fun.

As the success of the database depends on many users sharing their data, we aimed at making contributing data as intuitive and as easy as searching and visualizing data. We have thus simplified the data contribution process to a point where only very limited anatomical knowledge is needed, aiming at enabling physiologists without deep anatomical training to submit data as well.

Finally, to our knowledge the *IBdb* is the first database in the field of invertebrate neuroscience that combines functional data with anatomical data, and at multiple levels ranging from entire brains to single neurons. This ability, together with the possibility to deposit not only representative data but concrete sets of experiments, provides an opportunity for anatomists to directly interpret their findings in a functional context, as well as allowing physiologists to tether their findings to a coherent anatomical framework that automatically generates context for any functional data. The ease of use of the *IBdb*, combined with housing functional and anatomical data, has the potential to facilitate interactions between expert anatomists and physiologists and thereby strengthen structure-function analysis across the diversity of insect brains.

## Motivation to contribute

The landscape in the insect neuroscience research community has changed dramatically, and most relevant funding bodies are in the process of implementing open data mandates or have already done so (e.g. European Research Council, National Institutes of Health, Wellcome Trust, etc). Thus, the initial driving force for data deposition is much larger compared to that surrounding earlier database attempts. Yet, why should researchers use the *IBdb* for meeting these new mandates rather than other available databases? Different from open databases, for example Figshare.org, the *IBdb* is dedicated to insect neuroscience and thus provides all tools required to manage, annotate, and cross-link the specific data formats generated in this field. However, it is not rooted within a single laboratory, nor a single species, thus providing a framework for data from a broad research community. The unique possibility for cross-species comparison and the combination of anatomical and functional data additionally broadens the relevance of the *IBdb*.

Visualization of anatomical research data is often difficult and particularly complex when involving data from other research groups. We have implemented a range of tools enabling the visualization of data in fast, flexible, and effortless ways. This saves considerable time compared to other available software tools, in particular for complex 3D neuron data (e.g. Amira). The data contributed are also immediately incorporated into the framework of existing data. Outside the *IBdb* these data are distributed across many publications. Comparison of one's own data to any published data would entail contacting authors, obtaining files in unpredictable formats and finding ways to compare them to one's own work within the software a research group is currently using. The *IBdb* solves these issues and delivers such comparisons within seconds. Crucially, these advantages are already present immediately after data upload, prior to publication. Via the private mode of the *IBdb*, individual neurons can already be compared to their counterparts in other species while datasets are being obtained, enabling the user to generate visualizations suitable for conference contributions and publication figures. This possibility is to our knowledge unique to the *IBdb* and provides a major motivation for data contribution.

Importantly, the dual function of the *IBdb* as data repository and data management tool eliminates the need to reformat and prepare datasets for publication, a process that is required when submitting datasets to websites dedicated to only data deposition. The *IBdb* therefore provides a streamlined and integrated experience from data acquisition to publication, aimed a minimizing researcher workload. This is additionally relevant as formulating a data management plan has also

become mandatory for projects funded by most funding organizations. By using the *IBdb*, users in insect neuroscience not only have access to a dedicated data repository, but at the same time have a tool at hand for efficient handling of ongoing research data, making data management plans easy to design and effective. This is particularly the case when using the API for accessing the data, as up- and download can be automated and custom designed interfaces can be programmed according to the needs of any particular research group.

Independent from these aspects, the API of the *IBdb* offers users the opportunity to expose their published data directly to approved third party applications. The bundled availability of data from many research groups generates a possibility of data reuse with a much broader scope than any individual solutions, providing an increased motivation for developers to design workflows that incorporate deposited data. High-quality data deposited in the *IBdb* therefore additionally increases the visibility of the data owners.

## Biological insights

While the data currently deposited in the *IBdb* represents only a starting point to illustrate the functions and possibilities of our database concept and user interface, the neurons and species present in the *IBdb* already highlight new biological insights as well as concrete paths towards such insights.

Most obvious insights can be gained from data deposited in the *IBdb* that is not published or publishable elsewhere. For example, isolated neural morphologies from the Monarch butterfly have been published solely in the *IBdb* and demonstrated a previously unknown connection between the gall of the lateral complex and the mushroom body lobes (dp-GA-MB(L) neuron, NIN-0000383), thus directly linking the output of the central complex with the mushroom body for the first time in insects.

The possibility for direct comparison of neuropil and neuron data from many insect species has the potential to identify discrepancies in neuropil definitions across species and pinpoint solutions for revised homologies. For example, the main output neurons of the central complex (PFL or CPU1 neurons) are most likely homologous across insects and target the dorsal LAL in all species in which this region has been described (e.g. butterflies, moths, flies, beetles), except in locusts, in which they terminate in the ventral LAL. This discrepancy suggests that the definition of subregions of the LAL and their borders to surrounding brain regions (e.g. crepine) might have to be re-evaluated.

Along similar lines, the *IBdb* has generally the potential to expose discrepancies in functional data. Work carried out in a research group interested in sensory processing might define the function of a particular cell type very differently than a group focused on motor control, or one focused on neuromodulatory functions, simply because different experimental paradigms are aimed at different aspects of neural function and interpretations of results might be biased toward the underlying hypotheses. Such diverging views of neuron function can silently coexist in the literature for long times, but if bundled in side-by-side function entries on the same neuron profile page in the *IBdb* they become highly obvious. Such exposure will facilitate scientific discussion and the emergence of unified ideas of neural function.

Finally, in any field it is important to consistently define technical terms, agree on key concepts, and have consistent standards regarding what is acceptable data to validate conclusions. During the development phase of the *IBdb* it became clear that the isolated work in many insect model species has led to a wide range of cases in which similar terms have different meanings in different species and where views on what is acceptable proof for, for example, neural function varies. The definition of cell type is one such example, where researchers working in different species and brain regions attached substantially different meaning to the term. With the unifying approach of the *IBdb* we had to therefore identify common ground and define cell type in a way that was acceptable to each contributing party. Similar issues for other aspects of insect neuroscience research will likely be exposed by the increased levels of communication enabled by the *IBdb*. While resolving these issues will often take time and effort (and will involve researchers from across the field), the main function of the *IBdb* in this context is not to impose a strict set of rules, but to provide the framework in which inconsistencies can be exposed and resolved, eventually facilitating biological insight.

## Scalability and long-term sustainability

The *IBdb* is designed for long-term accumulation of data by many contributing research groups across the field of insect neuroscience. To successfully provide this service, the *IBdb* has to address several challenges of sustainability. These challenges are threefold: First, ensuring technical and financial maintenance of the database infrastructure; second, guaranteed, continued scientific oversight and expert curation, and third, lasting scientific relevance. We have therefore taken steps to address each of these points.

In the light of fast changing web technology, maintenance of high technical standards requires continuous effort and active updating of the database code, which in turn requires financial resources. The first issue is covered by an ongoing agreement with the web-developers that built and maintained the database software over the period of the last six years. Financially, voluntary contributions from research groups that initiated the database have paid for its creation. As the maintenance costs are a small fraction of the development costs, it will be easily possible to run the database within the framework of the existing service agreement for at least the next 5 years without any changes required. However, when the data volume increases substantially, the static costs of housing the data will increase accordingly. While keeping all public, persistent data available free of charge is mandatory (given that the *IBdb* functions as a public data repository), maintaining the *IBdb* as a free data management tool, that is allowing unlimited private data for each user, will likely become unsustainable over time. If this becomes a problem, free space in the private section of the database will have to be restricted. All space required beyond a certain limit will have to be rented to directly offset the costs for maintaining and administering these data. At the same time, research groups involved in creating the *IBdb* use this platform as their primary tool for management of research data and public data repository. The obligation to formulate data management plans and strategies for public deposition of research data, combined with the lack of equally suitable alternative platforms, will ensure that third party funding dedicated to maintaining the database will continuously be available via research funding of the founders of the *IBdb*.

To anticipate the slowly growing costs of housing the database due to increasing data volume, we aim at eventually relocating the data from the currently used commercial Amazon cloud platform to an academic server that is provided at minimal costs or free of charge. To this end we have ensured that the *IBdb* does not depend on any core functions of the Amazon cloud storage service, enabling to move the database to a new location with comparably moderate effort.

The second issue of continued quality control and expert curation is addressed at several levels. Firstly, at the level of species, the scientific administrator has the main responsibility to oversee the curation of the species entries by each species' dedicated group of curators. Given the limited number of species that can realistically be included in the *IBdb* over the coming years (our estimate is a maximum of several hundred), the associated workload for general oversight is limited and manageable by a single person. As species curators actively perform research on the species they are responsible for, there is a substantial self interest to maintain high standards to advertise their work and thus facilitate their research.

At the level of neuron entries, the workload is higher due to larger data volumes. Accordingly, the main responsibility for oversight is more distributed and lies with the species' curators. Once datasets exist in the database, updating is generally optional and will only in rare cases be necessary. In those cases, the strongest incentive for keeping data up to date lies with the data owners and research group leaders, who also possess the highest expertise for these data. Overall, expertise is mostly required to approve new datasets, a process requiring substantial overview over the data available for the relevant species. With an increasing number of species, the number of curators will also have to increase. This makes curator recruitment and training key requirements both for quality assurance and for enforcing consistent curation strategies. For this purpose, regular meetings of all curators will be held and approval of new curators will be carried out only after an extended training period. To attract new curators in the future, the *IBdb* administrators will actively approach researchers in the field of insect neuroanatomy. Finally, to identify existing quality issues with deposited data, establish routine workflows, to support curators and contributors at all levels, and to recruit new users, a dedicated, full time database curator position is being created, initially funded by members of the *IBdb* consortium for a minimum of one year.

Third, continued scientific relevance of the database will have to be ensured to attract users and contributors long term. Most crucially, after introducing the database to the field, the available deposited information has to grow beyond a critical point, at which the *IBdb* becomes the natural choice for depositing insect neuroscience data. We believe that this point has already been reached, as the number of deposited data sets is growing extensively (*Figure 7—figure supplement 1*) and users without affiliation to the founding consortium have begun to deposit data. Nevertheless, attracting more data will remain a key mechanism to increase the attractiveness and acceptance, and, consequently, the relevance of the database in the field. As one of the key advantages of our concept is the possibility to deposit all anatomical and functional data from insect neuroscience, irrespective of data format, we have begun to approach authors of published work to enable them to make old data available to the growing *IBdb* user base.

### *IBdb* usage for outreach and teaching

While highly useful for classroom teaching of structure-function relations in neural systems, the *IBdb* has proven to be invaluable to introduce new members of a research team to the basic layout of brains, neurons, and neural circuits in a particular species. Using the database serves as an easy (and fun) access point to available information on a research species, including key publications, and therefore saves significant effort when writing review papers, PhD thesis introductions, background sections for travel grants, etc. While this is true for established researchers as well, it is especially true for younger scientists and students who are new to the field or are at the beginning of their careers.

Finally, the *IBdb* provides the possibility for anyone to access original research data in intuitive and attractive ways (*Figure 7*). This provides opportunities to design teaching assignments for neuroscience students to carry out meta-analyses. With access to the data in the *IBdb* via the application programming interface (API), we have provided the possibility for third parties to develop dedicated teaching tools that provide streamlined methods to use the data for specific classroom exercises. Beyond researchers and students, journalists, interested members of the public, or members of funding bodies can also view and explore neuroscience data. Ideally, this will contribute to a more transparent understanding of what the output of science is and could spark increased interest in insect neuroscience.

### Widening the scope toward other animal groups

The framework we have generated with the *IBdb* is not limited to housing insect brain data. Without major modifications it would be equally suited for hosting data from other animal taxa. While the intuitive, schematic search engine would not be useful for comparing species that do not share a common basic brain outline (i.e. a relevant 'generic brain'), the text-based expanded search could allow the construction of queries across multiple groups, for example searches according to functional terms. We are currently conceptualizing the expansion of the database toward including spiders and envision that crustaceans and other arthropods would be logical next groups.

The *IBdb* therefore provides not only a tool for the insect neuroscience community to facilitate data management, data visualization, transparency of results and effective teaching, but it can easily be expanded toward related fields. Additionally, it might also serve as a blueprint for how to set up similar databases in unrelated research areas. In principle, the strategies used in the *IBdb* are applicable to any scientific field that can be linked to a hierarchical framework.

## Materials and methods

### Data location

All web infrastructure is hosted by Amazon web services on servers located in Frankfurt, Germany. Data is stored using a PostgreSQL relational database hosted by the Amazon Relational Database Service and files are stored using the Amazon S3 object storage service. The servers hosting the website and the local HANDLE system are running in Amazon EC2 containers, which runs Linux. Resources communicate using Amazon Virtual Private Cloud.

## Database framework

The database structure and interaction is managed by a python based Django application. User authentication, permissions, and data security are also managed within the Django application. A NGINX web server hosts static content and serves as a reverse proxy for dynamic content served by a uWSGI application server hosting the project's Django application. Asynchronous tasks are implemented using the Celery distributed task queue and RabbitMQ message broker.

Data is externally accessible via a web API delivering content in the JSON format to the front-end web application. The web API was implemented with Django and the Django REST framework.

Long-term data persistency is provided to allow users to reference information or profile pages on the site in scientific publications and other external media in a static state, while continuing to allow data to be updated as more information is acquired. When a request is made by a user for a persistent copy of a dataset to be created, a copy of the data related to the current state of the dataset is serialized and parsed into JSON. A persistent unique identifier is then assigned (HANDLE). The JSON data, HANDLE and additional metadata is recorded in a separate table and can no longer be modified. All files associated with the persistent dataset are marked as locked in the database and can no longer be modified by the user. The recorded state of that dataset can be accessed and viewed on the site using the url associated with the assigned HANDLE. The original data copied to create the persistent dataset can be modified without effecting the persistent dataset. Additional files may also be added, but will not be reflected in earlier persistent records.

## Graphical user interface

The front-end of the database is primarily implemented using the Angular web framework in Typescript, HTML and SASS (CSS extension language). The Typescript, HTML and SASS are compiled and bundled with the Angular CLI using WebPack to create the distributed application files targeting ECMAScript 2015 capable browsers. Graphical consistency is targeted for browsers using Webkit and Gecko-based layout engines adhering to web standards.

The web based three-dimensional viewer was implemented using Typescript, WebGL and the Three.js three-dimensional graphics framework. The two-dimensional schematic view, brain designer and path designer was implemented using Typescript, the Canvas API and Paper.js vector graphics scripting framework.

## Security measures

User to server communication is protected by the Hyper Text Transfer Protocol Secure (HTTPS). User authentication is managed by the Django authentication system. Access to data is restricted by object based permissions limited to authorized users through the web interface or API arbitrated by the Django server.

File downloads of protected content stored on Amazon S3 are accessed using time limited urls, assigned to an authorized user at the time of a download request by the Django server. Files are directly downloaded from the S3 storage to the user using the temporary URL. The process is seamless to the end user who needs only be logged into the website and click the link associate with the intended file. Downloads are logged and accessible to the owner of the data being accessed. Files can be uploaded to the S3 object storage only by authorized users with a time limited URL provided by the server. The upload is logged and associated with the contributing user. Publicly available thumbnail images and other reduced quality images are stored separately in a publicly accessible (read-only) S3 bucket.

Content explicitly designated as public is accessible through the graphical user interface or via the API to any authorized visitor. Private content is only accessible to authorized users given permission to access the data.

The PostgreSQL database is protected by a firewall allowing access only via the Amazon Virtual Private Cloud and is not open to direct access via the web.

## Nomenclature and brain hierarchy

All names for brain areas are in line with previous research. The brain regions of the generic insect brain follow the new insect brain nomenclature introduced for *Drosophila* by *Insect Brain Name Working Group et al., 2014*. Accordingly, we have established three hierarchical levels of brain

regions: super-regions, neuropils, and sub-regions. Super-regions are stereotypical and can be expected to comprise the ground pattern of the brain in all insects (although some might be reduced in certain species). The only exception to the *Insect Brain Name Working Group et al., 2014* scheme is the anterior optic tubercle, which we have raised to the level of super-region, given its prominence and distinct nature in most insect species. Sub-regions are often specific to individual species and therefore, if such regions were defined, we used the names given to them within the relevant species. We did not unify for example names of the mushroom body calyx divisions across species, as this would firstly imply homology where there might not be any, and second, novel naming schemes will have to be developed by the community and not be imposed by a data repository. Anticipating that changes to brain names can and will happen in the future, all names, as well as the level of a region within a hierarchy, can be modified.

Within some neuropils, regular, repeating elements can be found, usually defined as columns and layers. We have implemented such a system in the central complex, that is without having to define an array of sub-regions, several strata and orthogonal slices (following the new brain nomenclature) can be generated. The default number of slices in the generic brain is 16, assuming that this number is the ancestral state of this region.

Neuron names follow the conventions within each species, as there is no common naming scheme for insect brain neurons yet in place. However, we provide the possibility to define several alternative names for each cell type to allow the parallel use of names. This is possible as the identity of a neuron is linked to the persistent ID, and not to the neuron's name. Given that we house neurons from multiple species, we add a prefix to the full name of each cell type specifying the species, for example 'am' for *Apis mellifera*.

## Acknowledgements

We are indebted to the many test users of the *IBdb* who patiently located bugs and inconsistencies and thereby helped to streamline the database outline and make the user interface more intuitive. We also thank all members of the Heinze lab for many helpful discussions that improved the *IBdb* and this manuscript.

## Additional information

### Competing interests

Kevin Tedore: Kevin Tedore is a commercial web developer (founder and owner of Kevin Tedore Interactive) who designed and developed all software and interfaces underlying the insect brain database. The other authors declare that no competing interests exist.

### Funding

| Funder | Grant reference number | Author |
|---|---|---|
| H2020 European Research Council | 714599 | Stanley Heinze |
| H2020 European Research Council | 817535 | Marie Dacke |
| Air Force Office of Scientific Research | FA9550-14-1-0242 | Eric Warrant |
| Deutsche Forschungsgemeinschaft | EL784/1-1 | Basil el Jundi |
| Deutsche Forschungsgemeinschaft | HO 950/24-1 | Uwe Homberg |
| Deutsche Forschungsgemeinschaft | Me365/34 | Randolf Menzel |
| Julius-Maximilians-Universität Würzburg | | Keram Pfeiffer |
| Norwegian Research Council | 287052 | Bente G Berg |

| Freie Universität Berlin | | Randolf Menzel |
| --- | --- | --- |
| Swedish Research Council | 2014 - 04623 | Marie Dacke |
| Deutsche Forschungsge-meinschaft | HO 950/25-1 | Uwe Homberg |
| Deutsche Forschungsge-meinschaft | HO 950/26-1 | Uwe Homberg |
| Zukunftskolleg University of Konstanz | | Randolf Menzel |

The funders had no role in study design, data collection and interpretation, or the decision to submit the work for publication.

### Author contributions

Stanley Heinze, Conceptualization, Resources, Data curation, Funding acquisition, Validation, Visualization, Writing - original draft, Project administration, Writing - review and editing; Basil el Jundi, Conceptualization, Funding acquisition, Validation, Writing - review and editing; Bente G Berg, Data curation, Supervision, Funding acquisition, Writing - review and editing; Uwe Homberg, Randolf Menzel, Keram Pfeiffer, Data curation, Supervision, Funding acquisition, Validation, Writing - review and editing; Ronja Hensgen, Data curation, Validation, Visualization, Writing - review and editing; Frederick Zittrell, Data curation, Software, Validation, Visualization, Writing - review and editing; Marie Dacke, Data curation, Funding acquisition, Validation, Writing - review and editing; Eric Warrant, Funding acquisition, Writing - review and editing; Gerit Pfuhl, Jürgen Rybak, Data curation, Validation, Writing - review and editing; Kevin Tedore, Conceptualization, Resources, Data curation, Software, Validation, Visualization, Methodology, Writing - original draft, Project administration, Writing - review and editing

### Author ORCIDs

Stanley Heinze ⓘ https://orcid.org/0000-0002-8145-3348
Basil el Jundi ⓘ https://orcid.org/0000-0002-4539-6681
Bente G Berg ⓘ https://orcid.org/0000-0003-4035-9125
Uwe Homberg ⓘ https://orcid.org/0000-0002-8229-7236
Randolf Menzel ⓘ https://orcid.org/0000-0002-9576-039X
Keram Pfeiffer ⓘ https://orcid.org/0000-0001-5348-2304
Ronja Hensgen ⓘ https://orcid.org/0000-0002-4876-9084
Frederick Zittrell ⓘ http://orcid.org/0000-0001-7878-4325
Marie Dacke ⓘ https://orcid.org/0000-0001-6444-7483
Eric Warrant ⓘ https://orcid.org/0000-0001-7480-7016
Gerit Pfuhl ⓘ https://orcid.org/0000-0002-3271-6447
Jürgen Rybak ⓘ https://orcid.org/0000-0003-0571-9957
Kevin Tedore ⓘ http://orcid.org/0000-0002-2722-8393

### Decision letter and Author response

Decision letter https://doi.org/10.7554/eLife.65376.sa1
Author response https://doi.org/10.7554/eLife.65376.sa2

## Additional files

### Supplementary files

• Transparent reporting form

### Data availability

All data underlying the figures of the paper are freely available in the insect brain database: https://insectbraindb.org/. Access to the database is free and can be achieved either by browsing https://

insectbraindb.org/ or by API access. Documentation see https://insectbraindb.org/static/IBdb_User-guide.pdf.

The following datasets were generated:

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
