## [Decision Letter]

**Acceptance summary:**

This manuscript will be of interest to insect neuroscientists and broadly to the neuroanatomy community. It presents a new web resource that collects and displays neuron, brain region and species data in user-friendly ways. Ease of use and data (including functional) now on the site, contribute to fulfilling its potential as the insect neuroscience data hub.

**Decision letter after peer review:**

Thank you for submitting your article "The insect brain database – A unified platform to manage, share, and archive morphological and functional data" for consideration by *eLife*. Your article has been reviewed by 2 peer reviewers, and the evaluation has been overseen by Ronald Calabrese as the Senior and Reviewing Editor. The following individual involved in review of your submission has agreed to reveal their identity: Marta Costa (Reviewer #1).

Essential revisions:

1. A concern that must be addressed is the quality and effectiveness of the curation. The reviewers found that the curation of data was often incomplete or inconsistent. This might be a consequence of the distributed strategy for curation together with the significant direct input from users. What backstops for quality control and proofreading are in place?

2. The issue of cell type as defined in the database must be addressed. The authors use the term 'cell type' often in the manuscript, and a few times in the user guide. The concept of 'cell type' is an essential one for neuroscience researchers. However, we were unable to find any reference to it on the site, particularly in the information pages for neurons. This led to ambiguity about what data was displayed, what constitutes a cell type and how one would search for it.

3. Plans to incorporate (or not) *Drosophila* into the database by potentially by linking to, VirtualFlyBrain, and IBdb should be described and discussed.

4. Issues of sustainability should be addressed.

*Reviewer #1 (Recommendations for the authors):*

Below I list some questions for the authors that relate to each of the concerns listed in the public review.

1) Could the authors please clarify how is the assignment and naming of cell types done, and how that relates to the neuron name (full and alternate)? Furthermore, if there are additional cases of neurons being present twice (mirrored and non-mirrored) could the authors please make that clear on the neuron page, linking the 2 objects.

2) Could the authors please comment on what is their strategy to ensure accuracy, consistency and completeness of curation? The current section 'Curation and administration' does not provide enough information on this.

3) Could the authors please rewrite the highlighted sentence in the abstract for clarity?

*Reviewer #2 (Recommendations for the authors):*

1) As the authors mentioned, this comparative database does not include some species, most critically *Drosophila melanogaster*. This exclusion is a pity, as searching homologous neurons of the *Drosophila* neurons in other insects, or vice versa, would be inspiring and promote further comparative approach. As the same neuropil nomenclature was used in the largest and probably most elaborate database with similar functions for the *Drosophila* brain, VirtualFlyBrain, and IBdb, it would be helpful to implement cross-species neuron search based on arbor areas (as mentioned in Line 508).

2) More comprehensive 'preset' depository of published data would make this database more attractive, as users naturally tend to first go to the largest and most comprehensive one. VFB also made a big success in this respect by actively indexing massive data taken in different labs.

3) Maybe it is not a requirement of this article type, but I would have liked to see some demonstration of new biological findings using this new database.

4) There is a concern on sustainability, as administration/management (e.g. species, curation, approval) continuously need expertise. It would be powerful to come up with a mechanism to encourage participation of more active users.

5) This database doesn't seem to require registration of neurons to a standard brain of the species (Line 502). It is unclear how one can make visualization as in Figure 4 without registration. It would be helpful to detail what one can/cannot do depending on the data type.

6) Line 253 "To create a new species" It sounds like creationism. Better rephrase.

---

## [Author Response]

Essential revisions:1. A concern that must be addressed is the quality and effectiveness of the curation. The reviewers found that the curation of data was often incomplete or inconsistent. This might be a consequence of the distributed strategy for curation together with the significant direct input from users. What backstops for quality control and proofreading are in place?

This is a very good point and we have thoroughly thought about this issue. We have made several deep changes to the database and expanded the relevant section in the discussion:

1. As an initial note, many of the specific inconsistencies the reviewers have mentioned, in particular the duplicate datasets of mirrored neurons, were carried over from an early testing stage of the database, before our current standards were implemented. As all data is persistent, we cannot delete these datasets. However, as datasets becoming obsolete might be an issue that will emerge in the future as well, we have now implemented an archive function for neuron datasets. The persistent handle of this neuron will still exist and can be resolved, but the neuron will no longer appear in searches or lists, preventing cluttering of the database. We have updated the manual and the manuscript accordingly. Overall, this function now provides the curators with an additional tool to keep the database clean in the long run.

2. To address the quality of curation, we have now implemented another layer of approval: The curator approval. Every curator now has to go through a training phase during which she/he can approve datasets, but before this dataset goes live, it will have to be reviewed by the database administrator (lead author), enabling feedback on the curation process. Only once the curator has been sufficiently trained, she/he will be given full curation rights by the main curator.

3. To ensure more consistent standards between datasets, we have now designed checklists for data contributors and curators that make explicitly clear which data of each dataset is mandatory and how it should be provided. These documents are available as new items under the menu ‘Help’.

4. We have now added another checkpoint that prevents approval of incomplete datasets. To this aim, we have made several entries for each data category mandatory (neuron: arborizations, schematic neuron path, image, description, soma location [if unknown ‘unknown’ will be specified]; species: image, species description, schematic brain). Without valid data entered for these fields, no approval request can be sent to the curator.

5. A final step to increase data integrity has been added and concerns major updates of datasets. Whenever a new image stack, 3D reconstruction, or function entry is created (as well as when arborization regions are redefined), the re-publication process will now require curator review and re-approval. To this aim, we have redeveloped the new dataset management interface and added this tool to the top of each neuron page (“Record administration”; visible for owners and users with write-access to the entry).

2. The issue of cell type as defined in the database must be addressed. The authors use the term 'cell type' often in the manuscript, and a few times in the user guide. The concept of 'cell type' is an essential one for neuroscience researchers. However, we were unable to find any reference to it on the site, particularly in the information pages for neurons. This led to ambiguity about what data was displayed, what constitutes a cell type and how one would search for it.

We have added a cell type definition in the paper and on the database site, also in the user guide (“All neurons of one brain hemisphere that exhibit identical projection patterns are defined as belonging to the same cell type. In many cases a cell type will consist of one individual neuron, while in other cases, many identical neurons will comprise a cell type. In those cases, the first level of similarity beyond the individual neuron will be defined as the cell type.”). We have added a short paragraph about cell type definition to the results. We also added a new supplemental figure illustrating our cell type definitions with examples. Please find a more elaborate discussion of the topic here (a copy of which can also be found on the database forum):

We essentially pursue a connectivity based cell type definition in the long run, i.e. neurons that have the same up- and downstream connections belong to the same cell type (as in Hulse et al., 2020). Yet, while connectivity data is increasingly available, it does not exist for most species. Therefore, a first approximation is an identity defined by neural projection fields. As two neurons with identical projections can still have different connections, these projection based definitions will occasionally have to be subdivided into several cell types once connectomics data becomes available.

Several issues were considered in the process:

First, neuron types on the right and left hemisphere. While we assume that the insect brain is largely symmetric with respect to the midline, asymmetries are present and one cannot simply assume that all neurons from one side of the brain are completely identical to the other side (see asymmetrical body in the fly CX). To not obstruct discovering potential systematic differences between hemispheres (and for practical data handling issues), we define analogous neuron types from the right and left hemisphere as separate cell types. This is also in line with the fact that they originate from different neuroblasts, hence providing a larger degree of compatibility with lineage based cell type definitions.

Second, modular brain regions. In regions like the central complex and the optic lobe, highly similar neurons occur in isomorphic sets of repeating individuals. In the case of the central complex columnar cells, we classify those as different cell types (according to column identity), as it has become clear recently that the CX columns are not identical copies of each other, and they additionally derive from different neuroblasts. For the thousands of repeating modules in the optic lobe, this approach is not practical, as it would require that each columnar neuron (e.g. of the medulla) has a firmly assigned ommatidium. Additionally, while not all optic lobe cartridges can be expected to be identical, there will be a limited number of types (e.g. corresponding to ommatidial types). If those can be reliably identified, cell types can be assigned based on this information. Practically, the columnar processing systems in the optic lobe will be collapsed to a single column in the database, or to as many as required to capture the ommatidial diversity in a species.

Third, definitions of cell types across species: We define cell types only within single species. This is because a neuron could have the same function, but different developmental origin, the same origin but different connections, etc. While neurons with identical projection patterns will emerge when comparing across species, the thereby suggested homology will remain hypothetical unless developmental studies are conducted on these cells. This in fact highlights a major value of the insect brain database, i.e. the possibility to identify shared features of neural circuits across species and the development of novel hypotheses based on this information.

Highly organized neuropils: In neuropils like the mushroom body, antennal lobe, or the central complex, cell types can be embedded in a hierarchy, including supertypes, neuron families, and cell classes. As for species phylogeny, the same terms are likely not always representing the same level in the hierarchy (i.e. a supertype in one brain region might correspond to a class in another), but such terms nevertheless provide helpful guidance for classifying neurons in complex and organized brain regions (and probably in all regions). For example, in the CX, there are many types of columnar neurons, each defined by their specific projection patterns (e.g. all individuals of the PFNv-R3 type). Yet they can be grouped into several supertypes, defined by the projection patterns without referring to specific columns (e.g. PFNv), into neuron families, defined by the overall projection patterns (e.g. PFN), and into a neuron class (columnar CX neuron; as opposed to tangential neurons, or pontine neurons). We have added the possibility to add these higher level classifiers as tags to cell types, allowing more complex search queries and easier handling of increasing numbers of database entries.

3. Plans to incorporate (or not) *Drosophila* into the database by potentially by linking to , VirtualFlyBrain, and IBdb should be described and discussed.

We have now integrated *Drosophila* by cross linking our database to VFB as suggested by the reviewer. With the help of VFB staff we have mapped all brain regions in the IBdb to their corresponding neuropils in VFB, so that search results in IBdb (graphical search) have a direct corresponding search result in VFB. While searching in the IBdb, an API mediated query is sent to VFB and results are displayed in a newly introduced panel in the bottom half of the screen as a list of single neuron entries. Each list contains information obtained from VFB and is linked to the corresponding entry at VFB. This feature is available only for the graphic brain region search, but includes complex multi-neuropil searches as well. The feature is most useful for complex graphical search, e.g. showing all neurons connecting two or more neuropils (as numbers of neurons innervating individual neuropils are very large in Drosophila due to connectomics data). We moved the section describing this function from discussion to results (now that it is no longer just discussing a future direction but an existing tool). We updated the database manual accordingly.

4. Issues of sustainability should be addressed.

We see three points related to sustainability that are relevant for the IBdb long term

perspective:

Technical maintenance: This issue is essential to ensure persistence of deposited data. As discussed in the manuscript, we have planned the technical maintenance and the associated costs for the next 10 years and are confident that this plan is sustainable and suited to manage the site, keep existing data accessible and keep the code up to date regarding changing webstandards and data formats. In practical terms, this is facilitated by the fact that the lead author, as well as several co-authors, are using the site as primary data management and deposition tool for all ongoing and future research activities. The mandatory nature of data management and public deposition and the lack of suitable alternatives will ensure that third party funding dedicated to this purpose will continuously be available for technical maintenance, server fees etc.

Oversight to ensure quality of content: This issue requires continuous expertise, relevant for ensuring that new datasets meet all required standards and that old datasets are kept up to date, if new information becomes available. We believe that, at the level of cell types and experiments, the strongest incentive for keeping data up to date lies with the data owners, who also possess the highest expertise for these data. Overall, updating data is optional, and even without updates, any data deposited will adhere to the standards initially applied upon approval. The species curators, in coordination with the scientific administrator, have an oversight function to reinforce these standards (see point 1 above). As curators actively perform research on the species they curate, there is substantial self-interest to maintain high standards to facilitate their research. On the level of species, the scientific administrator has the main responsibility. Given the limited number of species that will be included (realistically not more than several hundred for the near future), the associated workload is limited and manageable by a single person. Additionally, we are actively pursuing dedicated funding for one to two full time assistant positions for database curation (a first position advert is being prepared). These positions will be devoted to performing routine maintenance, assisting new and existing users with data upload, identifying and resolving issues with existing datasets by regularly checking all entries, as well as actively attracting new users by identifying research papers with data appropriate for deposition in the IBdb. These assistants will also have the role to identify issues that regularly occur on the user end, with the aim of developing solutions to help streamlining and improving the site.

Finally, continued relevance of the IBdb: The initial incentives for users to deposit data are given by the visualization possibilities and exposure of the data. Additionally, the possibility to programmatically access own and other researchers’ data via API allows to integrate this data into automatic data analysis workflows, allow effective data management and increases the chance of data reuse by third party applications (e.g. computational modelling apps). These provide a strong, bottom up mechanism to ensure that data volume will increase over time. With sufficient data deposited, new users will more easily be attracted, continuously increasing the relevance of the database. Users without associations with any of the authors have already begun to deposit data on the site, so we believe that the critical mass of data has been reached to keep the site relevant and in long term use. We will facilitate this development by actively approaching authors of publications that produce data suited for deposition in the IBdb. To illustrate the momentum of the database usage, we have added a supplemental figure of the change of usage over time to provide a basis for extrapolation. We have updated and substantially expanded the sustainability section in the discussion and clarified the three aspects above.

Reviewer #1 (Recommendations for the authors):Below I list some questions for the authors that relate to each of the concerns listed in the public review.1) Could the authors please clarify how is the assignment and naming of cell types done, and how that relates to the neuron name (full and alternate)? Furthermore, if there are additional cases of neurons being present twice (mirrored and non-mirrored) could the authors please make that clear on the neuron page, linking the 2 objects.

We have added the cell type definition to the database, added a discussion of the topic to the forum, to allow public discussion. We also added a new figure to illustrate the principles of how cell types are defined in the database.

Neuron types: All neurons that cannot be distinguished based on their light-microscopy level morphology belong to the same cell type. (see response above for details) While many cell types will occur as single individuals in the insect brain, others will occur in many copies. For instance, there will be many individuals of Kenyon cells innervating the α/β lobes, but only single individuals of specific MB-output neurons. Corresponding neurons on different hemispheres are treated as different cell types in the IBdb, indicated by the automatically generated suffix ‘R’ or ‘L’ in the neuron name. A direct link to the contralateral cell type is now provided on the neuron page of each neuron (if the counterpart exists, e.g. for https://insectbraindb.org/app/neuron/61).

Neuron names: Neuron names follow largely the conventions from each species, and no

overall set of rules is enforced upon all neuron type entries in the IBdb. This ensures that neuron names do not have to change when deposited in the database, which might deter users. To make neurons more easily findable, when searched by name, we have included ‘alternative names’ as a search field that is automatically queried when searching for ‘name’. This alternative name can be any name by which this cell has been referred to in the literature, either in the same species or another one, if relevant (usually the *Drosophila* name is listed here if known). In general, the diversity of neuron names reflects the diversity of the field, and while a common naming scheme is highly desirable long term, it does not seem feasible to enforce such a scheme across species at this point. Nevertheless, we have introduced a species prefix and hemisphere suffix for each neuron type entry (both automatically generated), and recommend to use names of neurons that reflect their morphology, similar to what is used in the fruit fly. However, we realistically cannot hope that researchers change all historically used names for neurons, and think that this could even be counterproductive for continuity and findability of data in the field. Short names are defined in a way that allows researchers to effectively communicate about a neuron in publications or talks. We now have additionally provided optional entries for three hierarchical levels (super-types, neuron family, and cell class) that can be used to describe a neuron if needed (e.g. PFNv-R3 [cell type], PFNv [supertype], PFN [family], columnar CX neuron [class]).

Mirrored neurons: While useful for illustration purposes, artificially mirrored neurons do not necessarily reflect the true anatomy of the contralateral counterpart of a cell type, as hemisphere specific branching patterns might exist (even if very rare). We have therefore added an archive function to the database and have archived mirrored duplicates, so that they no longer show up in search results. As deleting public entries is not possible to ensure persistence, these neurons will still be reachable by the original handle, but the page will indicate that they have been archived. No duplicate neurons are present anymore.

We have updated the relevant section of the results in the manuscript.

2) Could the authors please comment on what is their strategy to ensure accuracy, consistency and completeness of curation? The current section 'Curation and administration' does not provide enough information on this.

Accuracy: We have now implemented an approval process for curators, who now can only approve neurons when being granted this right by the scientific administrator, after a training period. Datasets which are missing mandatory information can no longer be submitted for approval (mandatory information was expanded to now include not only arborizations, schematic neuron path and profile image, but also morphology description, soma location, and either a publication, or a note stating that there is no publication associated with that entry).

Completeness: There originally was no requirement for completeness in the database other than ensuring searchability and identifiability. We believe that this, in principle, is increasing the willingness for deposition of data by not demanding too much information from the owner at the time of deposition. Yet, we see the point that rudimentary datasets are not very useful and have developed a checklist for the data contributor that explains which information should be included. We added several data fields to the list of mandatory data (see above) and a request for approval will be blocked when these fields do not contain valid content.

We have updated and substantially expanded the relevant sections in the results and the discussion of the manuscript.

3) Could the authors please rewrite the highlighted sentence in the abstract for clarity?

While we do not know which sentence the reviewer refers to, we have completely revised the abstract in light of the revisions, and hope that this issue was resolved in the process.

Reviewer #2 (Recommendations for the authors):1) As the authors mentioned, this comparative database does not include some species, most critically *Drosophila melanogaster*. This exclusion is a pity, as searching homologous neurons of the *Drosophila* neurons in other insects, or vice versa, would be inspiring and promote further comparative approach. As the same neuropil nomenclature was used in the largest and probably most elaborate database with similar functions for the *Drosophila* brain, VirtualFlyBrain, and IBdb, it would be helpful to implement cross-species neuron search based on arbor areas (as mentioned in Line 508).

We completely agree and have now implemented a first version of cross links for search

results with VFB (see above for details).

2) More comprehensive 'preset' depository of published data would make this database more attractive, as users naturally tend to first go to the largest and most comprehensive one. VFB also made a big success in this respect by actively indexing massive data taken in different labs.

This is indeed an ongoing effort, however, the content of the database is not the primary

subject of the paper, but rather it is the novel structure and principles of user interaction. Nevertheless, we fully agree that a database framework without much content, or with many rudimentary datasets is not attractive and undermines the credibility of the claims we make. We have hence added much content based on our own research, both anatomical and functional. We have additionally added a ‘datasets’ item to the list menu that lists datasets corresponding to specific publications, increasing visibility for individual authors of these publications and allowing users to quickly locate neuron entries based on individual publications. We also added two large datasets of EM based projectomics work, introducing a new experiment type, the interactive experiment (with its own dedicated viewer). We also have modified the ‘specimen sex’ entry to allow depositing of not only male, female, unisex adult brains, but also brains of developmental stages. We are currently constructing entries

with the Tribolium (red flour beetle) community that will use this feature, further expanding the range of data that is present in the database.

3) Maybe it is not a requirement of this article type, but I would have liked to see some demonstration of new biological findings using this new database.

While not a requirement as such, we agree and have added unpublished neuron data from the Monarch butterfly, which is not part of any publication and likely will not become part of any paper. The most interesting neuron provides a novel direct connection between two brain regions (gall and mushroom body), for which no connection has been previously shown in any species. This illustrates that datasets in the IBdb can serve as independent means to disseminate knowledge (and data) through citable content.

We have also identified interspecies discrepancies in how parts of the LAL/CRE regions are defined that have become obvious only via the database. While not resolved yet, this example highlights how issues of brain region homology become apparent and can subsequently be solved with the aid of the IBdb. We have added a section about facilitating biological insights to the discussion that addresses insights that are based on the current content as well as ways in which the database facilitates such insight in principle.

4) There is a concern on sustainability, as administration/management (e.g. species, curation, approval) continuously need expertise. It would be powerful to come up with a mechanism to encourage participation of more active users.

Encouragement of users: We are planning to actively advertise the database via

presentations at (virtual and in person) conferences at the Arthropod Neuroscience Network and the International Society for Neuroethology, to encourage more users to contribute and identify users interested in active curation. As an incentive for curators, curation will increase visibility of the curator (as each curator is credited on each dataset they approved), which should be particularly attractive for early to mid-stage career researchers.

We see three points related to sustainability that are relevant for the IBdb long term

perspective:

Technical maintenance: This issue is essential to ensure persistence of deposited data. As discussed in the manuscript, we have planned the technical maintenance and the associated costs for the next 10 years and are confident that this plan is sustainable and suited to manage the site, keep existing data accessible and keep the code up to date regarding changing web standards and data formats. In practical terms, this is facilitated by the fact that the lead author, as well as several co-authors, are using the site as primary data management and deposition tool for all ongoing and future research activities. The mandatory nature of data management and public deposition and the lack of suitable alternatives will ensure that third party funding dedicated to this purpose will continuously be available for technical maintenance, server fees etc.

Oversight to ensure quality of content: This issue requires continuous expertise, relevant for ensuring that new datasets meet all required standards and that old datasets are kept up to date, if new information becomes available. We believe that, at the level of cell types and experiments, the strongest incentive for keeping data up to date lies with the data owners, who also possess the highest expertise for these data. Overall, updating data is optional, and even without updates, any data deposited will adhere to the standards initially applied upon approval. The species curators, in coordination with the scientific administrator, have an oversight function to reinforce these standards (see point 1 above). As curators actively perform research on the species they curate, there is substantial self interest to maintain high standards to facilitate their research. On the level of species, the scientific administrator has the main responsibility. Given the limited number of species that will be included (realistically not more than several hundred for the near future), the associated workload is limited and manageable by a single person. Additionally, we are actively pursuing dedicated funding for one to two full time assistant positions for database curation. These positions will be devoted to performing routine maintenance, assisting new and existing users with data upload, identifying and resolving issues with existing datasets by regularly checking all entries, as well as actively attracting new users by identifying research papers with data appropriate for deposition in the IBdb. These assistants will also have the role to identify issues that regularly occur on the user end, with the aim of developing solutions to help streamlining and improving the site.

Finally, continued relevance of the IBdb: The incentives for users to deposit data are given by the visualization possibilities and exposure of the data. Additionally, the possibility to programmatically access own and other researchers’ data via API allows to integrate this data into automatic data analysis workflows, allow effective data management and increases the chance of data reuse by third party applications (e.g. computational modelling apps). These provide a strong, bottom up mechanism to ensure that data volume will increase over time. With sufficient data deposited, new users will more easily be attracted, continuously increasing the relevance of the database. Users without associations with the authors have already begun to deposit data on the site, so we believe that the critical mass of data has been reached to keep the site relevant and in use long term. We will facilitate this development by actively approaching authors of publications that produce data suited for deposition in the IBdb. To illustrate the momentum of the database usage, we have added a supplemental figure of the change of usage over time to provide a basis for extrapolation.

5) This database doesn't seem to require registration of neurons to a standard brain of the species (Line 502). It is unclear how one can make visualization as in Figure 4 without registration. It would be helpful to detail what one can/cannot do depending on the data type.

We have now emphasized (in the paper, the database and the manual) that no shared 3D view can be carried out with neurons that are not registered into a common standard. While we encourage registration for more advanced visualizations, we have deliberately not made registration a requirement in order to allow easier deposition of older data and data from species without existing reference atlas. This limit is now also highlighted when uploading a neuron reconstruction that is not registered. This information was also highlighted in the user guide.

6) Line 253 "To create a new species" It sounds like creationism. Better rephrase.

Well spotted, we agree and have corrected this potentially misleading phrase to ‘To submit a new species…’